# *Helicobacter pylori* Related Diseases and Osteoporotic Fractures (Narrative Review)

**DOI:** 10.3390/jcm9103253

**Published:** 2020-10-12

**Authors:** Leon Fisher, Alexander Fisher, Paul N Smith

**Affiliations:** 1Department of Gastroenterology, Frankston Hospital, Peninsula Health, Melbourne 3199, Australia; 2Department of Geriatric Medicine, The Canberra Hospital, ACT Health, Canberra 2605, Australia; Alex.Fisher@act.gov.au; 3Department of Orthopedic Surgery, The Canberra Hospital, ACT Health, Canberra 2605, Australia; paul.smith@act.gov.au; 4Australian National University Medical School, Canberra 2605, Australia

**Keywords:** *Helicobacter pylori* infection, osteoporosis, fractures, falls, medications, management

## Abstract

Osteoporosis (OP) and osteoporotic fractures (OFs) are common multifactorial and heterogenic disorders of increasing incidence. *Helicobacter pylori* (*H.p.*) colonizes the stomach approximately in half of the world’s population, causes gastroduodenal diseases and is prevalent in numerous extra-digestive diseases known to be associated with OP/OF. The studies regarding relationship between *H.p.* infection (HPI) and OP/OFs are inconsistent. The current review summarizes the relevant literature on the potential role of HPI in OP, falls and OFs and highlights the reasons for controversies in the publications. In the first section, after a brief overview of HPI biological features, we analyze the studies evaluating the association of HPI and bone status. The second part includes data on the prevalence of OP/OFs in HPI-induced gastroduodenal diseases (peptic ulcer, chronic/atrophic gastritis and cancer) and the effects of acid-suppressive drugs. In the next section, we discuss the possible contribution of HPI-associated extra-digestive diseases and medications to OP/OF, focusing on conditions affecting both bone homeostasis and predisposing to falls. In the last section, we describe clinical implications of accumulated data on HPI as a co-factor of OP/OF and present a feasible five-step algorithm for OP/OF risk assessment and management in regard to HPI, emphasizing the importance of an integrative (but differentiated) holistic approach. Increased awareness about the consequences of HPI linked to OP/OF can aid early detection and management. Further research on the HPI–OP/OF relationship is needed to close current knowledge gaps and improve clinical management of both OP/OF and HPI-related disorders.

## 1. Introduction

Both *Helicobacter pylori* (*H.p*.) infection (HPI) and osteoporotic fractures (OFs) constitute major challenges for public health systems globally due to huge clinical and economic burdens. Accumulating evidence suggests that in health and disease stomach and gut directly and indirectly via multiple neurohormonal pathways regulate the musculoskeletal and other systems by controlling appetite, food intake, absorption of nutrients and energy balance [1,2,3,4,5,6,7]. These physiological relationships (including the gut/stomach–bone axis) may be affected by HPI. HPI is associated with numerous diseases in and outside the stomach, many of which have the potential to influence bone and muscle status, predispose to falls and, consequently, contribute to OFs. As *H.p*. colonizes the human stomach in over 50% of the world’s population [8,9,10,11] and HPI frequently coexists with OP/OF, deeper understanding the relationships among HPI-related diseases, the skeleton and falls becomes highly important; it may help to improve the preventive and therapeutic strategies for OFs. However, only a small number of studies have examined the association between HPI and bone status and the results have been controversial [12,13,14]. Although conventional wisdom suggests that in clinical practice the two main components of OFs—OP and falls—need to be integrated and viewed under complementary angles, the possible contribution of HPI-associated diseases to falls has not been addressed in the literature systematically.

Each year thousands of papers are published on HPI, OP and OFs. In this narrative review based mainly on the literature from the last decade, we present basic information on HPI, summarize the key findings and existing evidence in the literature for and against the impact of HPI-induced and -associated diseases on skeleton, falls and OFs, highlight the possible causes for controversies, discuss the practical implications of the accumulating knowledge and introduce a practical algorithm for management OP/OF incorporating the new data on the potential role of HPI.

## 2. Brief Overview of *Helicobacter pylori* Infection

*H.p.* is a spiral-shaped, flagellated, microaerophilic, Gram-negative bacterium, which coevolved with humans >50,000 years [9,15]. The bacterium, discovered in 1982 by Warren and Marshall [16,17], colonizes the stomach in approximately 4.4 billion individuals [10,18] and is currently recognized as the most important microbiological agent in human upper gastrointestinal tract disorders. The prevalence of HPI (about 30% in developed countries and up to 80% in developing countries) varies by age (higher in the elderly, especially among institutionalized people); socioeconomic, urbanization and sanitation conditions; lifestyle and diet factors; and geographical regions (Central/South America, Asia, Eastern and Southern Europe have 50–80% higher prevalence than the rest of the world) [19]. In the past decades, the HPI rates declined in developed countries but remained high in rest of the world.

The long co-existence of *H.p.* with humans resulted in high level of genetic diversity and extensive polymorphism (especially among strains from different ethnic and geographic origins [20]), multiple strategies and complex mechanisms of colonization and persistence, ability to maintain a mild inflammation of the gastric epithelium and escape from and/or attenuate host immune system response (Figure 1).

Although the gastric mucosa is well protected against infection, *H.p.*, compared to most bacteria and viruses, is well adapted to survive in acidic conditions of the human stomach. It produces urease, an enzyme which hydrolyses urea causing the pH rise (essential for stomach colonization), and gamma-glutamyl transpeptidase, an enzyme which supports its growth and survival in the gastric mucosa. Acute HPI affects also the parietal cell proton pump mechanism, increases production of cytokines and activates neural pathways that stimulate somatostatin and inhibit both histamine production and acid secretion [21,22,23,24]. Furthermore, *H.p.* alters the mucus barrier by modulating the expression of stomach mucins [25]. These mechanisms counteract the acidic environment of the stomach (first defense line) and play a key role in *H.p.* survival and colonization. Urease, in addition to its role in acid neutralization, contributes to *H.p.* pathogenicity by production ammonia (disrupts cell junctions and damages epithelium) and reactive oxygen species (ROS), activating lipoxygenase, inducing angiogenesis, hypoxia-induced factor and apoptosis [26,27,28,29,30]. The helical shape and flagella, two factors responsible for bacterial mobility, also contribute to colonization and persistence of the infection (allow *H.p.* to escape low gastric pH by moving to the “protective” mucus layer before colonizing the gastric epithelium).

*H.p.* strains (the microbe encompasses approximately 1600 genes) have different genes encoding virulence factors (encoded proteins) which are secreted, membrane-associated or translocated into cytosol of the host cells via the IV type secretion system, where they can affect the host cell functions. The most studied virulence factors implicated in the pathogenicity of *H.p.* are produced by strains containing the following genes [29,31]: cytotoxin-associated gene A (cagA), vacuolating cytotoxin gene (vacA), duodenal ulcer (DU) promoting gene (dupA), induced by contact with epithelium gene (iceA), blood group antigen-binding adhesin (babA), sialic acid binding adhesin (sabA), outer inflammatory protein A (oipA), adherence-associated lipoprotein A and B (alpA/B), *H.p.* outer membrane protein Q (hopQ), gamma-glutamyl transpeptidase (GGT) and high-temperature requiring protein (HtrA). The regulation and function of the proteins encoded by these genes are complex processes. Gastric acidity is detected by the *H.p.* and serves as a trigger to increase production of pathogenic factors needed to subvert host defense [32]. *H.p.* strains carrying virulence factors are considered more pathogenic compare to the strains lacking these factors.

H.p. induces activation of most components of innate (epithelial, neutrophil, macrophage and dendritic cells) and adaptive immunity (B and T cells) [30,33,34,35]. H.p., especially with specific virulent strains, in addition to a local reaction results in systemic low-grade inflammation [36,37,38]. Exosomes (extracellular phospholipid vesicles formed by exocytosis) containing CagA and other virulence factors, can be distributed by the circulation and might be involved in the development of extragastric disorders [39,40,41,42]. Furthermore, molecular cross-mimicry between H.p. components and gastric H/K-ATPase, platelet surface, endothelial, fibroblast and smooth muscle cells as well as atherosclerotic plaque antigens may trigger immune and inflammatory responses and tissue destruction [11,42,43,44]. Noticeable, *H.p.* has the ability to induce and modulate host’s immuno-inflammatory responses and to protect itself by blunting host’s ability to eliminate the bacteria: the long-standing inflammatory environment is harmful to the host but tolerant for the microbe (favors persistence).

With regard to OP, CagA is the only *H.p.* virulence factor evaluated so far. CagA is a 125–140 kDa protein encoded in the complex of the cytotoxin associated gene pathogenicity island (cag PAI) and delivered into gastric epithelial cells via bacterial type IV secretion system [45,46,47]. The C-terminal region of CagA contains multiple Glu-Pro-Ile-Tyr-Ala (EPIYA) motifs. Various combinations of four different EPIYA segments (A–D) create structural polymorphism that enables classification of individual CagA into subtypes; the two major subtypes are the Western and the East Asian (Japan, China and Korea). Almost all CagA carry EPIYA-A and -B; EPIYA-C is prevalent in Western countries (ABC-type CagA), while EPIYA-D -is prevalent in East Asian countries (ABD-type CagA). CagA possesses also a 16-amino-acid sequence termed CagA multimerization (CM) sequence, two CM in the Western CagA type and one CM in the East Asian type [48,49,50]. CagA interacts with 25 host cell receptors in the gastric epithelium demonstrating the highest quantity of all known virulence-associated effector proteins in the microbial world [51]. CagA-positivity has been shown to be associated with increased production of proinflammatory (IL-1β, IL-6, IL-8, IL-17, TNF-α, IFN-γ and CRP) and anti-inflammatory (IL-4 and IL-10) cytokines, suppressed phagocytosis, induced tolerogenic dendritic cells and blocked T cell responses [52,53,54]. *H.p.* strains that express CagA often also express other virulence factors (e.g., VacA, BabA, etc.); such strains tend to influence more host immune responses. In Western countries only 30–40% of strains have cagA, whereas in East Asian countries up to 100% of strains carry cagA [55,56,57,58,59,60,61,62]. The serological immune response to H.p. cagA+ strains in various geographic regions of the world ranges from 32% to 80% [63,64,65], with the lowest prevalence in Europe and highest in Asian countries.

CagA plays a critical role in the development of peptic ulcer disease (PUD), chronic gastritis, gastric cancer and mucosa-associated lymphoid tissue (MALT) B-cell lymphoma [41,65,66,67], as well as many extra-gastroduodenal diseases [68]. In Western countries, subjects infected with cagA-positive (cagA+) strains compared to those infected with cagA-negative (cagA−) strains of *H.p.* are at a 1.7–2.8-times higher risks of both PUD or gastric cancer [57,67,69], whereas, in the East Asian population, in which most *H.p.* strains carry cagA gene, these risks are 2.8–4.6-times higher [67]. The interactions between various bacterial virulence factors and their contribution to the immune and clinical phenotype of HPI outcome are complex; for instance, vacA polymorphism was found as one of the most important factors associated with anti-CagA-IgG seropositivity [65].

*H.p.* is acquired during childhood (within the first 5–10 years of life) via oral–oral and fecal–oral routes or contaminated water and unless eradicated by treatment persists for life. The specific virulence factors (abovementioned proteins encoded by the *H.p.* genes) are associated with microorganism’s survival/adaptation, anatomical distribution and pathogenicity (including effects on host immune and inflammatory responses). The variety of HPI clinical manifestations (development of different HPI-related diseases or asymptomatic survival) is also significantly influenced by the host genetic polymorphisms including cytokine gene polymorphisms [33,70,71,72]. As in the majority of subjects the immune system is unable to eradicate the HPI, acute gastritis is followed by chronic gastritis (histologic gastritis is present in all individuals with HPI) in one of three forms: (1) corpus/fundus-predominant; (2) antral-predominant; and (3) diffuse. The topographic distribution of chronic HPI-induced gastritis is at least partly host specific [8].

Although most (70–80%) infected individuals are asymptomatic, HPI is etiologically associated with acute gastric inflammation, chronic non-atrophic and atrophic gastritis, PUD (approximately in 10%), gastric adenocarcinoma (in 1–3%) and MALToma (in <0.1%) [73,74,75,76,77]. Moreover, HPI in addition to diverse gastroduodenal pathologies has been reported to be associated with multiple extra-digestive disorders, including cardiovascular, neurological, hematological, endocrine and other diseases known to be linked to OP/OF. However, some diseases (allergic, autoimmune and metabolic) are observed more frequently in subjects free of *H.p.* [9,75,78,79] suggesting that HPI may have a potentially “protective” effect.

In chronic HPI, the physiologically tightly regulated gastric acid, pepsin and hormonal (peptides and amines) secretion status is determined by the predominantly affected anatomic site. HPI-induced predominantly antral gastritis decreases the D-cell numbers and production of the inhibitory peptide somatostatin by these cells, increases production of gastrin (by antral G-cells), acid (by parietal cells), and pepsin (by chief cells), predisposing to duodenal ulcer disease. Corpus/fundus infection is associated with loss of parietal, chief and endocrine type cells, various degrees of atrophic gastritis and intestinal metaplasia resulting in reduced acid and pepsin secretion, lower levels of ghrelin (a pleiotropic hormone synthesized and secreted mainly by fundic P/D1 cells in humans and by gastric X/A-like endocrine cells in rodents) and histamine production (by the enterochromaffin-like cells, ECL) and elevated gastrin and gastric leptin secretion; these predispose to gastric ulceration and/or gastric cancer but protect against duodenal ulceration and, probably, against acid-induced gastroesophageal reflux disease (GERD) [58,80,81,82,83,84]. The changes in gastric structure and function are caused by products of H.p. itself and/or upregulated expression of proinflammatory cytokines released in response to HPI, especially with more virulent strains (e.g., cagA+, VacA, etc.).

Of note, *H.p.* has been detected in numerous sites throughout the body, beyond the stomach, including the oral and nasal cavities, gall bladder, large intestine, liver, coronary arteries, trabeculum, iris and skin [85,86]; emerging data indicate that the oral cavity can act as an extragastric reservoir of HPI [87,88,89], although some researchers oppose this conclusion [90,91].

In sum, the outcome of HPI is determined by the interactions of: (1) HPI virulence factors; (2) hosts’ age, gender and immune-inflammatory response (dependent on both host’s genetic susceptibility/resistance and *H.p.* characteristics); and (3) environmental and lifestyle factors (diet, cigarette smoking, alcohol consumption, sanitation and air pollution) [8,9,60,67,92,93,94]. Although colonization with *H.p.* is not a disease in itself, the condition significantly increases the risk of developing various clinical disorders—gastroduodenal and extra-digestive. This short overview provides background information on the complexity of factors involved in the pathogenesis of HPI-related diseases: genetic and geographic heterogeneity of the bacterium, the diversity of molecular mechanisms responsible for its persistence and effects in the human stomach, differences in host’s susceptibility and role of socioeconomic, sociodemographic, environmental, cultural and lifestyle factors (Figure 1). Understandable, these aspects as main determinants of HPI clinical outcomes should be taken into account when the relationship between H.p. (potentially commensal/symbiotic bacteria) and any clinical disorder is studied. Precise knowledge of HPI virulent factors, host susceptibility and environmental conditions are essential to understand and explain the variety of HPI clinical effects, including the potential HPI–OP/OF link.

## 3. HPI and Bone Status

Data on the relationship between HPI and OP are limited and controversial. Most studies were retrospective, single-center, cross-sectional, non-randomized (Table 1). Two recent meta-analyses came to opposite conclusions [12,14]. The first analysis [12], which included five studies involving 1321 participants (1068 women), found no significant association between HPI and OP (pooled OR 1.49, 95% CI 0.88–2.55, *p* = 0.14). In the second report [14], based on 21 observational studies with 9655 participants, HPI was significantly associated with OP (OR 1.39, 95% CI 1.13–1.71, *p* < 0.001). The results of these two meta-analyses should be interpreted cautiously in light of heterogeneity (stated by the authors of both reports) and obvious limitations: no information on *H.p.* virulence factors, host’s inflammatory/immunological responses and environmental factors, often small sample sizes, as well as lack of data on main risk factors for OP and no adjustment for such potential confounders as site, type and severity of gastric pathology; comorbidities; race/ethnicity; and pharmacotherapy used.

We identified 20 original studies (duplicated publications excluded) on the topic; these included in total 38,558 subjects (38,497 adults and 61 adolescents). Six studies included only women, mainly postmenopausal; in one report, the gender of the participants was not mentioned [95], in the rest of the studies there were 5801 males and 5522 females (M:F ratio 1:1.05). Of the 20 studies, 19 had a cross-sectional design, one analyzed a prospective cohort [96]. Nine studies were performed in Western countries (Italy, Turkey, Brazil (two reports from each) and Iran (three reports)) and eleven studies in Eastern countries/regions (Japan (five reports), Taiwan (four), South Korea (one) and China (one)).

A positive relationship between HPI and osteoporotic characteristics was demonstrated in nine studies, mainly from Eastern countries/regions (Japan (three reports), Taiwan (three reports) and Korea (one report)), and in two studies from Italy. In contrast, among the 11 studies which did not find such association, seven were from Western countries (Turkey, Brazil (two reports from each) and Iran (three reports)) and four from Eastern countries/regions (Japan (two reports), Taiwan (one) and China (one)). In other words, an association between HPI and bone mineral density (BMD) status has been observed in seven of eleven studies undertaken in Eastern countries and only in two of nine studies from Western countries. In one study, HPI was associated with BMD only at the lumbar spine but not at the femoral neck [97] indicating that the trabecular bone may be more affected than the cortical bone. The influence of HPI on bone health (and other organs) is not dependent only on the presence of the infection itself but is closely related to microbe’s virulence factors. Therefore, association with geography is to be expected, as Asian populations almost invariably have cagA+ *H.p.* strains.

Among the HPI virulent factors only the cagA genotype was evaluated in two Italian studies [98,99]. In the first of these studies (*n* = 240 men), 51 of 80 (63.7%) patients with OP and 107 of 160 (66.8%) non-OP controls were seropositive for HPI, but individuals infected by a cagA+ strain compared to a cagA− demonstrated significantly increased bone resorption (as defined by higher amounts of urinary cross-laps), reduced estrogen levels but similar BMD [98]. The second study (*n* = 1118, including 935 women and 183 men) presented the following additional data supporting the notion that cagA+ *H.p.* should be considered as a risk factor for OP and fractures in both genders: (1) a significant negative association between anti-CagA antibody titer and BMD; (2) a higher prevalence of cagA+ *H.p.* in osteoporotic (30%) and osteopenic (26%) patients compared to subjects with normal BMD (21%), although the overall HPI prevalence in the three groups did not differ significantly (41.5%, 46.2% and 43.9%, respectively); (3) only 30% of females and 14% of males with anti-CagA antibody titer above the median level had a normal BMD; and (4) hip and symptomatic vertebral fractures occurred in 4% of cagA+, in 2% of cagA− patients and in 0.8% of uninfected subjects (*p* < 0.05 for cagA+ vs. uninfected individuals) [99].

Other researchers, unfortunately, did not differentiate patients by H.p. virulence strains. However, findings in seven of eleven studies from Eastern countries indicate that prevalence of OP is approximately two-fold higher among individuals with HPI; in multivariate logistic regression analyses (after adjusting for age, gender, body mass index (BMI) and use of proton pump inhibitors (PPIs)), odds ratios (ORs) for OP ranged between 1.62 and 5.33 (Table 1). In this regard, it is interesting to compare geographic differences in cagA+ prevalence and variations in fracture epidemiology. In the last two decades, hip fracture rates are declining in Northern Europe, North America and Oceania [100] and increasing in Asian countries (Japan, Korea, Singapore and Lebanon) [101], where HPI prevalence is higher and most H.p. strains are cagA+; the rates of vertebral fractures in Asia are also among the highest in the world [102]. It was projected that in Southeast Asian countries in 2050 compared to 2018 hip fracture rates may increase 2.8–5.6-fold [103,104]. The differences in *H.p.* virulence, host and environmental factors in West and East may, at least partially, explain the role of cagA+ *H.p.* in OP/OF.

The relationships between HPI and bone turnover markers were addressed in three studies [96,98,105]. In adolescents, HPI was not accompanied by significant changes in serum levels of bone formation markers (N-terminal cross-links of human procollagen type I (P1NP), N-mid-osteocalcin (OC) and bone-specific alkaline phosphatase (bALP)), bone resorption marker (β-collagen I carboxy terminal telopeptide (β-CTX)), calcium and phosphate, as well as in circulating estradiol, intact parathyroid hormone (PTH) and ferritin levels, but a tendency to increased bone resorption (as reflected by higher β-CTX levels, *p* = 0.063) was observed [105]. In this study, the decreased vitamin B12 level was the only parameter differentiating the groups with and without HPI. Consistent with these findings are results from a prospective cohort study on postmenopausal women [96]: 5.8 years of follow-up revealed no significant differences between the *H.p.* seropositive and seronegative subjects in BMD and age-adjusted bone turnover markers (osteoprotegerin (OPG), receptor activator of nuclear factor kappa B ligand (RANKL), the RANKL/OPG ratio, OC and cross-laps); the differences between groups in BMD and serum OPG levels observed on univariate analysis were lost after adjusting for age. In logistic regression analyses, both *H.p.* and *Chlamydia pneumoniae* seropositivities did not predict incident lumbar spine or femoral neck OP [96].

The influence of HPI eradication therapy on BMD has been reported in two studies. The Japanese study (*n* = 255) found that *H.p.* seropositivity was an independent and significant risk factor for OP (OR 3.00, 95% CI 1.31–6.88, *p* = 0.009), whereas the success of *H.p.* eradication was not related to OP; use of PPIs was associated with a tendency (*p* = 0.073) towards OP [106]. Data from a large Taiwan’s National Health Insurance Research Database (5447 patients with PUD treated for HPI) demonstrated that early microbe eradication (within one year of being diagnosed) reduced the effect of infection on development of OP when the follow-up period was above 5 years [95].

The complexity of HPI–host interaction may contribute to the discrepancies observed in the reported studies. The many contradictory results could be due to abovementioned variations in multiple factors influencing the HPI outcome including the microbe’s specific virulence constituents (e.g., cagA and vacA); host’s demographic, genetic and clinical parameters; and environmental characteristics; these factors in fact were not considered in most of the studies. Obviously, the analysis of HPI–OP/OF association needs to be more complex.

To summarize, the published results, though mixed, emphasize the possible role of HPI, particularly with cagA+ strains, in OP and fractures. A positive correlation between presence of HPI and OP has been observed in Western populations infected with *H.p.* strains containing cagA and in 7 of 11 studies from Eastern countries where near 100% of *H.p.* strains possess cagA. These observations indicate a possible strain-specific association between HPI and OP: the odds of OP in patients with HPI appear to be about twice as great in those infected with cagA+ strains. Moreover, a potentially preventive effect of HPI eradication has also been reported. However, because of the mentioned above methodological weaknesses of the available studies an accurate and reliable conclusion on causal relationships could not be made.

## 4. HPI-Induced Upper Gut Diseases and Osteoporotic Fractures

Another important piece of information on the HPI–OP/OF relationship could be expected from data regarding HPI-associated chronic diseases, many of which as well as the medications used are known to have deleterious effects on bone metabolism and/or increase risk of falls contributing to OFs. In the following sections, we summarize the available information on potential links between chronic HPI-related gastroduodenal and extra-digestive tract diseases and OP/OFs.

Although only a small fraction of patients with HPI develop peptic ulcer disease (PUD), chronic/atrophic gastritis and malignancies, these diseases are common worldwide, and HPI is the most important etiological and pathogenic factor for their development. These disorders and/or concomitant treatment have been reported to be associated with OP/OF.

### 4.1. Peptic Ulcer Disease

HPI is responsible for 95% of duodenal ulcers and 85% of gastric ulcers which usually arise at the junction of the antral and corpus mucosa. At least eight studies showed that PUD is an independent risk factor (RF) for OF [110,119,120,121,122,123,124,125]. History of PUD has been found to be associated with an increased risk of osteoporotic thoracic vertebral fracture in a large population sample of Finnish men (*n* = 30,000) but not in the women [119]. In a Polish study of 240 females [120], women with PUD (*n* = 143, mean age 60.3 years) not using hormone replacement therapy (HRT) had lower BMD in all studied regions as compared to controls without PUD (*n* = 120, mean age 58.4 years); moreover, among HRT users, the BMD in lumbar vertebrae and Ward’s triangle was also significantly lower in women with PUD, whereas calcium intake was similar in both groups. Two reports from USA demonstrated an association between PUD and periprosthetic fractures after total hip replacement (*n* = 14,065; hazard ratio (HR) 1.5, 95% CI 1.1–2.2) [122] and total knee arthroplasty (*n* = 17,633; HR 1.87, 95% CI 1.28–2.75) [121]. In the Japanese study (*n* = 200, 105 women), multivariate analysis revealed that PUD (OR 4.98; 95% CI 1.51–16.45), along with HPI (OR 5.33; 95% CI 1.73–16.42) as well as common RFs (age, female gender, BMI), was independently related to OP [110]. A population-based study from Taiwan [123], which included 27,132 patients (aged ≥18 years) diagnosed with PUD and 27,132 randomly selected subjects (age- and gender-matched) without PUD, found that the OP risk (adjusted for covariates) was 1.85 times greater in the PUD group (13.99 vs. 5.80 per 1000 person-years). The highest risk was observed one year after PUD diagnosis (HR 63.4, 95% CI 28.2–142.7); use of a PPI significantly increased the OP risk (HR 1.17, 95% CI 1.03–1.34). Consistent with these data are results from two most recent large South Korean studies [124,125]. In the first prospective study (*n* = 10,030), PUD patients demonstrated a significantly higher OP risk (men: HR 1.72, 95% CI 1.02–2.92; women: HR 1.62, 95% CI 1.20–2.18); OP developed in 29.9% women and 11.1% men with PUD vs. 16.5% and 4.8% in controls, respectively [124]. The second report [125], based on analysis of 50,002 patients with PUD and equal number of controls matched by age, gender, past medical history, income and residence region, found increased risk of OP in PUD regardless of gender (adjusted HR 1.36, 95% CI 1.33–1.40). In contrast, a retrospective cross-sectional study from China (*n* = 867, with PUD 351 patient) reported that PUD was significantly associated with decreased BMD only in univariate analysis (OR 1.37, 95% CI 1.03–1.82) [117]. Lastly, in a cohort of patients operated for PUD between 1956 and 1985 (pre-HPI era) and followed for 30 years, the risk of OF was significantly (and independently of surgical procedure type) increased showing a standardized incidence ratio of 2.5 for the proximal femur, 4.7 for vertebra and 2.2 for the distal forearm [126].

Although these reports, as any observational study, cannot indicate causality, the relationship between PUD and fragility fractures is suggested. It appears that PUD may approximately double the risk of OP/OF.

### 4.2. Chronic/Atrophic Gastritis

It is well established that HPI is responsible for the majority (>90%) of chronic/atrophic gastritis [127,128] and plays an important role in the initiation of autoimmune atrophic gastritis [128,129,130]; the latter occurs in 2% of the general population with a higher prevalence in older (>60 years) females [131]. CagA+ *H.p.*, especially the East-Asian type, compared to the CagA− type induces more severe gastritis and mucosal atrophy and is more closely associated with gastric cancer [132]. Simultaneous presence of the CagA and other virulence factors (VacA, Helicobacter cysteine-rich protein C and the chaperonin Gro) increases the risk of chronic atrophic gastritis (a precursor lesion to gastric cancer) 18-fold [133].

Two publications by a Brazilian group reported that in postmenopausal women neither HPI, no atrophic chronic gastritis were associated with BMD or OP [107,108]. No association between HPI-induced chronic gastritis and OP was also observed in an Iranian study [134]. In contrast, a Norwegian study [135] found that, in patients with chronic atrophic gastritis, compared to sex- and age-matched controls, bone formation markers (OC, sclerostin, OPG and OPG/RANKL ratio) were lower and the incidence of OP was higher (the latter abnormality only in males). In older Korean women, presence of atrophic gastritis was significantly linked to OP after adjusting for seven variables including age, BMI, metabolic and lifestyle variables (OR 1.89, 95% CI 1.15–3.11) [136]. Similarly, Japanese men with HPI-induced atrophic gastritis (defined by serum pepsinogen I and pepsinogen II levels) demonstrated an increased risk of low trabecular bone density (OR 1.83, 95% CI 1.04–3.2) [112]. In a small Norwegian study (*n* = 17 patients, 41 controls), subjects with chronic atrophic gastritis, compared to controls, have: decreased circulating levels of OC (bone formation marker), sclerostin (an inhibitor of bone formation), osteoprotegerin (OPG) and OPG/RANKL ratio; unaffected levels of P1NP (bone formation marker) and bCTX (bone resorption marker); and (in males only) lower lumbar BMD and increased frequency of osteopenia and OP. No difference in bone quality assessed by microidentation was found [135]. These features were interpreted as suggestive of decreased bone formation and higher bone resorption in patients with chronic atrophic gastritis. A study from Germany reported that OP development was associated with gastritis/duodenitis (OR 1.14; *p* = 0.045) and PPI use [137]. A retrospective cohort study of Korean premenopausal women in their 40s (*n* = 983) who had undergone a 48-month follow-up assessment of BMD of L1–4 showed that atrophic gastritis (diagnosed by gastroduodenoscopy) was significantly associated with bone loss (adjusted for confounding factors); patients with persistent atrophic gastritis exhibited a greater decrease in BMD and the prolonged duration of the disease correlated positively with the amount of BMD reduction [138]. Atrophic gastritis and CagA seropositivity were associated with lower hemoglobin levels, and anemia was 2.6-times (in women) and 1.5-times (in men) more common among persons with atrophic gastritis [139]. Severe hypochlorhydria or achlorhydria were found in 44% of patients with idiopathic iron deficient anemia and in 1.8% among healthy controls [140].

Molecular mimicry between *H.p.* antigens and gastric H/K-ATPase has been proposed as a mechanism responsible for the association between HPI and development of chronic atrophic autoimmune gastritis [127,130,141,142,143,144]. In 20–30% of patients with HPI, autoantibodies to the H/K-ATPase were identified. In this organ-specific autoimmune disorder, autoantibodies to gastric parietal cells (in 90% of patients) and intrinsic factor (in 70% of patients) cause gastric gland atrophy, achlorhydria and hypergastrinemia (which induces hyperplasia of the ECL cells) resulting in vitamin B12 and iron malabsorption/deficiency and leading to megaloblastic/pernicious anemia and/or iron-deficient anemia, respectively [145,146,147,148,149,150]. The disease is clinically heterogeneous and may have an asymptomatic course. An inverse correlation between *H.p.* density and vitamin B12 levels has been shown [151]. Low serum B12 levels affect DNA synthesis, an important factor for bone remodeling. Vitamin B12 (and other B vitamins—B2, B6 and folate—linked to homocysteine metabolism) is regarded an essential factor for bone health [117,152,153,154,155,156,157,158,159,160,161]. Vitamin B12 deficiency was reported to have an increased fracture risk: 1.7- to [162] 1.9-fold [163] for hip fracture, 1.8-fold for vertebral fracture [163] and 2.9-fold for distal forearm fracture [163]. Peripheral neuropathy, occurring in vitamin B12 deficient patients [164], undoubtedly, increases risk of falls. Reversal of severe OP associated with pernicious anemia has been demonstrated after vitamin B12 replacement combined with etidronate (an antiresorptive bisphosphonate) therapy [165]. Repletion of B12 resulted in an 80% reduction in hip fracture risk among stroke patients [155]. However, recent trials and a meta-analysis did not show a preventive effect of treatment with vitamin B12 and folic acid on fracture risk [166]. Daily supplementation with B vitamins did not affect markers of bone turnover and did not reduce fracture risk in middle-aged and older women at high risk of cardiovascular disease [167]. Interestingly, a significantly increased hip fracture risk persists years after correction the vitamin B12 deficiency, indicating the independent pathophysiological importance of chronic atrophic gastritis and achlorhydria [162]. Chronic atrophic autoimmune gastritis is associated with multiple other nutritional deficiencies, including calcium, vitamins D, C and folic acid, each of which may affect the skeletal, nervous and hematological systems [149,157,168]. In addition, autoimmune gastritis clusters with autoimmune thyroiditis and type 1 diabetes mellitus [128,130], conditions linked to OP/OF. Iron deficiency with or without anemia has also been recognized as a RF for OP/OFs in many [169,170,171,172,173,174,175,176] but not all [177,178] studies (the topic is discussed in following sections).

Importantly, pharmacologic HPI eradication results in gradual and significant improvement in chronic atrophic gastritis [179,180,181]. On the other hand, when interpreting the studies on HPI-induced chronic/atrophic corpus gastritis, it has to be kept in mind that with progression of the severity and extension of atrophic lesions *H.p.* is spontaneously eradicated [143,182,183].

In conclusion, findings in patients with HPI-induced (especially with cagA+ strains) chronic/atrophic gastritis suggest that gastric corpus structural and functional (e.g., hypoacidity, hormonal disbalance, etc.) changes and associated nutritional deficiencies may negatively affect bone metabolism, neuromuscular and a wide range of other functions predisposing to OP, falls and OFs.

### 4.3. Gastric Cancer

HPI is an important determinant of neoplastic gastric lesions classified by WHO/IARC [184] as class 1 human carcinogen for non-cardia gastric adenocarcinomas [185,186,187]. HPI increases the cancer risk 5.8–7.9-fold [185,188], and the risk is 2–3-times higher in subjects infected with cagA+ strains [186,189,190,191].

According to most but not all studies [192,193], gastric cancer survivors who underwent gastrectomy, compared to the general population [194,195,196,197,198,199,200,201,202,203,204] or age- and sex-matched healthy controls [205,206,207,208], have significantly lower BMD, higher prevalence of osteopenia/OP (38.3% [196] to 55% [203]) and higher fracture rates (approximately 40% [196,200,205]). Bone loss (although of a lesser degree) was also reported in gastric cancer survivors after endoscopic tumor resection undertaken in early stage [202]. In South Korea, nationwide cohort study of cancer survivors who underwent gastrectomy (*n* = 133,179 matched to non-cancer controls, 1:1) demonstrated an increased risk of fractures (HR 1.61; 95% CI 1.53–1.70), which was higher in patients after total gastrectomy (HR 2.18; 95% CI 1.96–2.44) and adjuvant chemotherapy (HR 2.01; 95% CI 1.81–2.23); the elevated OF risk was significantly associated with anemia [208]. In a report from Japan, the adjusted hazard ratio for OF in men after gastrectomy (*n* = 132) was 2.55 (95% CI 1.17–5.55) and 3.56 (95% CI 1.33–9.52) in those who survived >20 years [209].

A considerable amount of OFs after gastrectomy occurs in the early postoperative period [200,203]. Bone remodeling imbalance with disproportionately increased bone resorption [197,199], decreased BMD [210] and higher fracture rates [200] were often observed during the first postoperative year. Altered bone metabolism was reflected by increased serum concentrations of bone resorption markers (C-terminal telopeptides of type I collagen, deoxypyridinoline and pyridinoline) [199], elevated serum PTH and alkaline phosphatase (ALP) levels [197,210] and associated with vitamin D deficiency [210,211,212,213,214]. Some researchers, however, observed no changes in BMD, a slight elevation of OC and only minor increase in PTH levels after total gastrectomy [192]. The causes of OP and subsequent fractures in patients with gastric cancer are multifactorial. Malabsorption (especially of calcium, phosphate, iron, proteins, vitamins B12 and D), malnutrition, weight loss, use of certain medications (e.g., fluorouracil and cisplatin which induce apoptosis of osteoblasts and increase osteoclast activity [215,216,217], hormones, radiotherapy, comorbidities, physical inactivity, old age and smoking–all were documented as factors contributing to bone loss and OFs in gastric cancer patients [198,200,202,203,214,217,218].

It is well established that HPI eradication reduces the incidence of gastric cancer [219,220,221,222] and favors regression of the low-grade B-cell gastric MALT [223].

### 4.4. Gastroesophageal Reflux Disease (GERD)

HPI does not influence the function of the lower esophageal sphincter, the motility of the esophagus and the esophageal acid exposure. HPI might protect the distal esophagus (possibly an evolutionary adaptation [9]) by causing atrophy of the fundal gastric glands and hypochlorhydria, especially in subjects with cagA+, vacA+ strains and pro-inflammatory genotypes (IL-1β and IL-1RN) [224,225,226,227,228,229]. Many studies, as would be expected, reported an inverse association between HPI-induced corpus gastritis and GERD, its severity, prevalence of Barrett’s esophagus (BO) and esophageal adenocarcinoma [9,137,230,231,232,233,234,235,236,237,238]. The strongest relationship was observed in East Asian populations [229,237]. A meta-analysis of 72 studies (84,717 patients with BO and 390,749 controls) found that HPI reduces the risk of BO by 32% (OR 0.68, 95% CI 0.58–0.79) [237]. Six meta-analyses on association of HPI and esophageal adenocarcinoma indicated an inverse relationship [238], whereas a recent meta-analysis (35 studies including 345,886 patients) did not find such association, except the Middle East data [239]. Other researchers concluded that presence of HPI might aggravate GERD [240], or, at least, is not “protective” against GERD, as the incidence of GERD and its sequelae in patients with HPI is higher than that after eradication of the infection [241,242], HPI eradication improves GERD symptoms and esophagitis [243,244,245,246,247] and does not increase the risk of BO [248,249,250]. Meta-analyses on effect of eradication HPI produced, however, inconsistent results [229]: a significantly higher risk of developing de novo GERD was demonstrated in Asian studies [251], but not in Western ones [252,253,254]. In a recent retrospective large cohort study from US (*n* = 36,803 patients with HPI), rates of esophageal and proximal gastric cancers 5, 10 and 15 years after treatment/eradication of HPI were low—0.15%, 0.26% and 0.34%, respectively [255]. In the interpretation of the data on the relationship between HPI and GERD the type and location of HPI-induced gastritis should be taken into account. As the level of gastric acid secretion is the main pathophysiological factor in GERD, chronic atrophic corpus gastritis causing hypo-/achlorhydria may exert a “protective” effect, while antrum gastritis with hyperchlorhydria can play an opposite role, and, not unexpectedly, HPI eradication may differently affect outcomes.

Several studies reported an association between GERD and vertebral fractures or kyphosis [256,257,258]. The most recent publications, however, did not confirm that GERD and decreased BMD are linked [117], neither that the incidence of OFs is higher among subjects with BO [259] (Kumar S 2017). As in the total population, older age, female gender and a higher comorbidity index were the independent risk factors for OFs in patients with BO. In the BO cohort, PPI therapy even prolonged and in high-doses, was not associated with increased fracture risk (HR 0.89; 95% CI 0.12–6.55), although a predisposition (numerically but non-significantly) for osteoporotic hip and vertebral fractures was observed [259].

The Maastricht IV/Florence Consensus Report on the management of HPI acknowledges that GERD is less common amongst those who are infected, but concludes that eradication of H.p. does not influence the severity of GERD [260]. In patients with GERD, according to Italian guidelines [261] and other recommendations [247], HPI can be eradicated.

### 4.5. Effects of Acid-Suppressive Drugs

Since gastric acid-lowering drugs (proton pump inhibitors (PPIs) or histamine-2 receptor antagonists (H2Rs)) are prescribed routinely in HPI-induced diseases (the “gold standard” therapy in acid-related disorders) and widely used in many other disorders (GERD, BO, prevention of aspirin- and NSAID-induced upper gastrointestinal bleeding, etc.), clarification of their effect on OP/OF is important. The topic remains a matter of debate. A positive, albeit modest, association between PPIs and OP was reported in animal [262,263,264] and numerous of human studies. Many studies and meta-analyses suggested that acid inhibitors, especially the PPIs, moderately increase risk of fractures, particularly in older adults (who are already at higher fracture risk); the risk increases with longer durations of PPI use, often only in the presence of at least one other RF for OP such as older age, female gender and a higher comorbidity score [265,266,267,268,269,270,271,272,273,274,275,276,277,278,279,280,281,282,283,284,285,286,287,288,289,290]. Moreover, use of PPIs may also increase risk of falls [291,292,293]. These associations, however, have not been confirmed in a number of longitudinal studies and reviews [259,280,294,295,296,297,298,299,300,301,302,303,304,305,306,307,308,309]; even a modest reduction in fracture risk with PPI use has been reported [307,310]. Some researchers who are not supporting the association of PPI therapy with BMD recognize, nevertheless, that in PPI users the risk of fractures and falls could be higher [311,312] as the unadjusted HR was significantly elevated (1.36, 95% CI 1.19–1.55).

In a recently published meta-analysis [313] which included 33 studies (*n* = 2,714,502), subjects taking PPIs demonstrated a significantly increased overall fracture incidence (22.04% vs. 15.57% in controls) with pooling OR of 1.28 (95% CI 1.22–1.35); fracture risk raised with duration of PPI use (OR 1.29 in short users and 1.62 in long users), but no effect on BMD was found. Similar results were obtained in another meta-analysis [289]: PPI users, compared to non-users, had an increased risk of developing spine fracture (HR 1.49; 95% CI 1.31–1.68), hip fracture (HR 1.22; 95% CI 1.15–1.31), any-site fracture (HR 1.30; 95% CI 1.16–1.45) and OP (HR 1.23; 95% CI 1.06–1.42), but there was no correlation with BMD loss neither in the spine, nor in the hip. According to the latest meta-analysis (24 observational studies with 2,103,800 participants, 319,568 hip fracture patients), risk of hip fracture increased significantly in PPI users (RR 1.20, 95% CI 1.14–1.28), and the association was observed even in subjects taken low doses of PPI [314].

It is noteworthy that in patients with chronic diseases, long term PPIs use may also contribute to muscle alterations such as muscle wasting, function loss and sarcopenia [315], which have been explained by nutrient deficiencies (magnesium, vitamin D) and gut dysbiosis related to acid suppression.

Regarding the effects of H2RA, most investigators observed little influence on BMD even in long-term (>5 years) users [268,269,279,290,301,314], but some researchers found lower BMD (if calcium intake was reduced) [316] and an increased hip fracture risk with high doses [276].

When interpreting the above mentioned controversial reports it should be acknowledged that an overlap between HPI status and using PPIs or H2RAs are important but often unrecognized confounders; in most of these studies the effects of other potential confounders (comorbidities, risk of falling, calcium and vitamin D supplementation, use of osteoporotic drugs, lifestyle factors, physical activity, etc.) have rarely been taken into account.

There is no general consensus on the clinical significance of possible detrimental effects of gastric acid-lowering drugs on the bone metabolism, BMD, falls and OFs. The acid-suppressive medications have been considered as a factor contributing to OFs by the American National Osteoporosis Foundation [317,318], while the American Gastroenterological Association (AGA) did not recommend routine screening or monitoring of BMD in PPI users [280].

No randomized controlled studies on the topic have been published until 2019. The first randomized, double-blind, placebo-controlled trial [319] that assessed in 115 healthy postmenopausal women (mean age 62 years) skeletal effects of 26 week treatment with a PPI found small but significant increases in circulating bone turnover markers: P1NP (+18.2% on esomeprazole 40 mg/day and +19.2% on dexlansoprazole 60 mg/day) and CTX (+22.0% and +27.4%, respectively); no statistically significant changes in serum PTH, fractional calcium absorption, as well as in urine levels of minerals and—most importantly—no impact on areal BMD in the lumbar spine, total hip, or femoral neck were observed. A large randomized double-blind trial of patients (*n* = 17,598) receiving pantoprazole (*n* = 8791) or placebo (*n* = 8807), rivaroxaban alone, rivaroxaban with aspirin, or only aspirin demonstrated that three years of PPI use was associated only with an excess of enteric infections but not with fractures or other previously reported adverse events (cardiovascular, pneumonias) [320]. In these two randomized studies the HPI status, unfortunately, has not been mentioned/evaluated.

When balancing the risks and benefits of long-term using PPIs it should be taken into consideration that several studies reported a significant relationship between the risk of gastric cancer and long-term use of PPIs [321,322,323,324,325,326,327,328,329]. In a recent large population-based study (*n* = 63,000, median follow-up of 7.6 years) a dose- and duration-dependent relationship between long-term use of PPIs and gastric cancer development, even after successful eradication of *H.p.*, has been shown (HR 2.44, 95% CI 1.42–4.20) [325]. A meta-analysis of reports on 926 386 participants obtained similar results [328]. At the same time experimental studies demonstrated that PPIs increase the effectiveness of chemotherapy for gastric cancer and might play a “dual role” in gastric carcinogenesis and its management [330].

To summarize, although the two randomized controlled studies did not find an association between PPI therapy and OP/OF, the potential risk of chronic PPI use for OF (and other adverse effects) observed in many previous studies and shown again in recent meta-analyses should not be neglected, particularly in persons with an elevated risk for OP and/or falls in whom the indications and the benefit/harm ratio should be carefully evaluated and lowest doses of PPIs for the shortest duration need to be prescribed.

## 5. HPI-Associated Chronic Extra-Gastroduodenal Diseases, Medication Use and Osteoporotic Fractures

HPI is one of few conditions that, aside from the common upper gut diseases, present high degree of concomitance with numerous extra-gastroduodenal disorders. A possible association between HPI and over 50 systemic diseases has been described. In recent years numerous reviews focused on this topic [11,13,14,68,79,331,332,333,334,335,336,337,338,339,340,341,342,343,344,345,346]. Most of HPI–linked extra-gastroduodenal chronic diseases and disorders as well as treatments used for are known to increase risk of OP, falls and OFs, especially in the elderly [317,347,348,349]. More than 80% of patients with a clinical vertebral or non-vertebral fracture suffer at least one chronic disease, and in 65.5% of these patients at least one bone- and/or fall-related RF was found [350]. Although the morbidity associated with OP is primarily due to fragility fractures, the factors contributing to OP and falls often are studied separately and the potential combined impact of HPI on these two main and equally important components of OFs remains largely unknown. There is no comprehensive synthesis of the data on involvement HPI in these processes. HPI-associated diseases are currently not recognized as RFs for OP/OF and HPI treatment is not integrated in prevention strategies.

### 5.1. Chronic Extra-Gastroduodenal Diseases

The presence of chronic diseases and use of medications that have deleterious effects on bone metabolism resulting in low bone mass, microarchitectural alterations and leading to fragility fractures are usually defined as RFs for “secondary” OP (in contrast to “primary” OP which is age-related and occurs in post-menopausal women and in men in the absence of an underlying disease) [351,352,353,354,355,356,357,358,359].

The main HPI-related chronic extra-gastroduodenal diseases and disorders relevant (but not all firmly established) concerning OP, falls and OF are listed in Table 2. To date, solid data support the etiological role of HPI in few disorders. In addition to the gastroduodenal diseases (described in the previous section), only hematological disorders—iron deficiency, vitamin B12 deficiency and immune thrombocytopenia (ITP)—are included in the international consensus (Maastricht V/Florence) and management guidelines recommend eradication of the pathogen [360]. Although many studies reported that HPI was associated with an approximately two-fold increased risk of development of other chronic extra-gastroduodenal diseases listed in Table 2, the topic remains controversial. Among 13 autoimmune diseases evidence in support of a probable role of HPI was found in four—TP, Graves’ disease, neuromyelitis optica and psoriasis [361]. Just to mention some examples of contradictory conclusions: many reports and meta-analyses suggested a positive correlation between HPI, metabolic associated fatty liver disease (MAFLD; previous term non-alcoholic fatty liver disease, NAFLD) [362,363,364,365,366,367,368,369,370,371,372], type 2 diabetes mellitus (DM)/insulin resistance [373,374,375,376,377], diabetic complications [378,379,380] and obesity [381,382], whereas others did not found such associations [383,384,385,386,387,388,389] and some described an inverse correlation of HPI with obesity [390,391,392]. Similarly, there is discrepancy between various studies regarding links between HPI and cardiovascular diseases (CVD) [333,393,394,395,396,397,398,399,400]. However, a recent large cohort study demonstrated that HPI eradication (in patients <65 years old) was associated with a trend of decrease in coronary artery disease (CAD) occurrence and significantly lower mortality rate [399]. Although studies suggesting the influence of HPI on chronic extra-gastric diseases pathogenesis are accumulating, the available data do not allow unequivocal conclusions.

Notable, certain *H.p.* strain-specific virulence constituents, in particular cagA, are significantly associated not only with an increased risk of gastroduodenal diseases but also with a number of extra-intestinal disorders and diseases associated with OP/OF, including iron deficient anemia, ITP, acute coronary syndromes, serum dyslipidemia, Parkinson’s disease, MAFLD, metabolic syndrome, DM, thyroiditis, liver cirrhosis and glaucoma. CagA activates major signaling pathways regulating bone remodeling (e.g., NF-kB, Wnt/beta-catenin, etc.) and alters various cellular responses involved in systemic inflammation, tissue homeostasis and function. Among diverse and complex pathophysiological mechanism through which *H.p.* cagA+ strains may cause extra-gastroduodenal disorders persistent systemic inflammation and molecular mimicry of CagA appear as most important.

Recent data suggests that OP and a variety of HPI-associated chronic extra-gastroduodenal diseases share several RFs, many aspects of pathophysiology and are RFs for one another. The common RFs and pathogenic mechanisms include malnutrition, weight loss, low grade systemic inflammation, oxidative stress, vitamin D insufficiency/deficiency, lower socioeconomic status, behavioral and lifestyle characteristics (cigarette smoking, excess alcohol consumption, lower physical activity, air pollution). Comparison of the role of sex and age in OP/OF and in HPI is of interest. OP/OF affects more females in the older/postmenopausal age (the F:M ratio among hip fracture patients is 2:1 [401]) and the incidence of OFs increases exponentially with age >70 years in both genders [402,403,404,405], while males <40 years of age had higher fall rates [406]. Among adults, a small male predominance of HPI-related outcomes has been found in a meta-analysis based on 169 studies (OR 1.12, 95% CI: 1.09, 1.15) [407]. Shared genetic basis underlying OP and a variety of HPI-related chronic extra-gastroduodenal diseases (CAD, DM, dyslipidemia) has been reported [408,409,410,411,412,413,414,415,416,417,418,419]. The shared biology and bi-(multi-)directional links between OP and HPI-associated chronic diseases apply particularly to CVD, CKD, CLD, DM and neurodegenerative diseases; these disorders are interrelated and often accompanied by OP/OF [420,421,422,423,424,425,426,427,428,429,430], and the risk of OP/OF increases when two or more HPI-associated chronic diseases are present.

On the other hand, it should be acknowledged that HPI may have a dual role of in human pathology. By affecting immune and inflammatory responses [78,92,431,432,433]. HPI may protect against asthma and allergy (particularly in children), autoimmune disorders (systemic lupus erythematosus, rheumatoid arthritis [RA] and multiple sclerosis, coeliac disease), inflammatory bowel disease (especially Crohn’s disease), eosinophilic esophagitis, eczema, obesity [9,78,342,433,434,435,436,437,438,439,440,441] and tooth loss [442]. Negative associations between these diseases and HPI (beneficial effects) were observed mainly in patients colonized with cagA+ strains of *H.p.* The inverse correlations were interpreted as a reflection of evolutionary adaptation [9] and/or improved socioeconomic conditions, sanitation and widespread use of antimicrobials. However, the topic is still debated. HPI has been reported to increase the risk of adult-onset asthma [443] and systemic lupus erythematosus [444]. A significantly higher HPI incidence rate was found in patients with Sjogren syndrome [445], autoimmune thyroid disease [446,447,448], multiple sclerosis [447,448] and no association between HPI and body mass index/obesity was observed in different populations [388,449,450]. The huge heterogeneity of asthma and other mentioned above diseases may, at least partially, contribute to the controversies in the publications.

### 5.2. Falls

Increasing evidence indicates that HPI may play a role in the complex and multifactorial nature of falls. Because OP is only part of the fracture equation, of particular importance is the fact that many mentioned above HPI-associated chronic diseases and their various combinations affect simultaneously (directly and/or indirectly) bone homeostasis and muscle mass/function, altering mobility, gait and balance, causing hemodynamic instability and collectively elevating the risk of falls and fractures, especially in the elderly [451,452,453,454,455,456,457,458,459,460,461,462,463,464,465]. It has been estimated that 87% of all fractures in the elderly are the result of a fall and 5–10% of all falls result in a fracture [466,467,468]. Among 494,160 patients aged ≥50 years with OP, 9% had only OP, while the rest had also chronic diseases, including CVD (54%), DM (8%), depression (4%), COPD (1%), with two or more diseases in 24% [469]; the HPI status in this study has not been reported.

OF is usually a result of a low trauma fall in a person with frail bones (osteopenia/osteoporosis), but up to two-thirds of all fractures are not attributable to OP if the latter is defined by BMD measurement [470]. Although muscle atrophy and bone loss may occur simultaneously, bone fractures are often preceded by loss of muscle mass and strength. The endocrine functions of skeletal muscles [471,472,473,474,475,476,477], the tight bidirectional (patho) physiological muscle-bone cross-talk [142,209,478,479,480,481,482,483,484,485,486,487,488,489,490,491,492], and their shared genetics [493,494,495,496] are well recognized. Not surprisingly, RFs for OP and falls often coexist (up to 63% in patients with HF [497]). Osteosarcopenia (loss of lean body mass, bone and muscle), a major component of aging-related illnesses, especially those linked to chronic inflammatory state, is common in many HPI-associated diseases and disorders (congestive heart failure [CHF], CKD, COPD, DM, RA, stroke, dementia, malnutrition, altered vitamin and hormonal status) and increases with ageing [485,491,498,499,500,501,502,503]. Postural control is impaired in women with low BMD [504]. A recent meta-analysis (5 cohort studies, *n* = 27,990) demonstrated a significant positive association between sarcopenia and fracture (adjusted HR 1.50, 95% CI 1.08–2.08) [505]. Increased falls and fracture risks were reported in patients with combination of sarcopenia and OP [483,506,507], but, surprisingly, a synergistic effect has not been observed in community-dwelling older men [508]. Noticeable, RFs for falls linked with HPI-related morbidity include, in addition to altered muscle status, impaired vision, hemodynamic instability (orthostatic and postprandial hypotension), arrhythmias, mental impairment, depression and anxiety [457,466,468,509,510], as well as the numerous medications used. Strong associations between frailty and an increased propensity to falls, fractures and mortality are well documented [483,511,512,513,514,515,516,517,518,519,520]. A vicious cycle may occur: chronic HPI-associated diseases contribute to frailty which is a significant determinant of OP/OF and disability/frailty further worsened after the fracture.

### 5.3. Medications

Among numerous drugs used for HPI-related diseases many are well known to contribute to OP/OFs. In regard to OP, these include corticosteroids, antidepressants (especially, selective serotonin- and serotonin-norepinephrine reuptake inhibitors), glitazones, opioids, benzodiazepines, antipsychotics, antiparkinsonian drugs, antiepileptics, PPIs, H2RA, thyroxine, furosemide, aromatase inhibitors, gonadotropin releasing hormone (GnRH) agonists, whereas hormone replacement therapy with estrogen, thiazides, angiotensin-converting enzyme (ACE) inhibitors and angiotensin II receptor blockers (ARBs), spironolactone, beta-blockers, statins, antihistamines, metformin, sulphonylureas, glucagon-like peptide-1 receptor agonists (liraglutide) and nitrates have shown an osteoprotective effect [316,521,522,523,524,525,526,527,528,529,530,531,532,533,534,535,536,537,538,539,540,541,542,543,544,545,546,547,548,549,550,551,552,553,554].

The prevalence of suspected medication-related falls is about 41% [555] to 49% [556]. Increased risk of falls was documented in users of anxiolytics/hypnotics, opioids, sedatives, antihypertensives (especially alpha-blockers), antidepressants, antiparkinsonian medications, antiepileptics and antiarrhythmics [557,558,559,560,561,562,563,564,565,566,567,568,569,570,571,572,573,574,575,576,577,578,579,580,581,582,583]; polypharmacy is strongly associated with injurious falls and fractures [574,582,584,585,586,587,588].

Notable, some medications demonstrate opposite effects on bone metabolism and risk of falling. For instance, thiazide diuretics, beta-blockers, calcium-channel blockers and ACE inhibitors may contribute toward orthostatic hypotension, syncope and falls, but exert beneficial effect on mineral-bone metabolism and, paradoxically, may reduce fracture risk [547,552,589,590,591,592,593,594,595,596,597,598]. Some agents have shown divergent effects on bone and skeletal muscles. For example, thiazolidinediones demonstrate detrimental effect on the skeleton and increase fracture risk [545,599,600,601,602,603,604] but a beneficial effect on muscle atrophy [605]; other anti-diabetic drugs (sulfonylureas, metformin and possible incretin mimetics) have a neutral or a positive/protective effect on bone health, but they may increase propensity for falls through hypoglycemia (insulin and sulfonylureas) [601,603,604,606]. When analyzing the complex relationships between OFs and drugs used it should also be taken into account that many medications (corticosteroids, sulfonamides, urea derivatives, vitamin K antagonists, cardiac glycosides, loop diuretics, potassium-sparing diuretics, ACE inhibitors, serotonin reuptake inhibitors, calcium-channel blockers and antiepileptic drugs) may affect the vitamin D status and calcium homeostasis [607,608,609,610,611]. Importantly, even in diseases inversely associated with HPI the abovementioned drugs may contribute to OP and/or falls and should be used with caution, especially in individuals with high fracture risk.

In other words, while the HPI-associated diseases and their complications may itself play an important role in OP/OF, certain concomitant treatment/medications may affect in the same or different way the skeleton, muscles and hemodynamic status, thereby modulating fracture risk. Prevention medication-related OP/OFs involves multifaceted concerns, and reviewing, ceasing, switching or dose reduction of prescribed drugs need to be considered. Obviously, the impact of each HPI-related disorder and the benefit/harm balance of drugs used should be evaluated in combination to determine an individualized preventive and therapeutic approach to OP/OF.

In closing this section, it worth mentioning that beneficial effects of HPI eradication include decreased occurrence and/or improvement across an array of chronic diseases and disorders associated with OP/OFs (e.g., iron deficiency/anemia, ITP, CAD, Parkinson’s, endocrine, psoriasis, gut dysbiosis, chorioretinopathy, etc.). HPI eradication is accompanied by positive effects on energy homeostasis, mineral and bone metabolism, reduces risk of muscle wasting and improves bioavailability of different orally administered drugs.

To sum up, HPI-associated chronic diseases via complex and possible shared underlying pathophysiological and genetic mechanisms and treatment-related factors may affect the musculoskeletal and other extra-digestive systems and therefore increase the risk of falls and fractures and vice versa OP/OFs exacerbate the chronic illnesses. Considering that HPI may contribute, at least partially, to many diseases associated with secondary OP, which occurs in up to two thirds of men, more than half of pre- and peri-menopausal women and in 20–30% of postmenopausal women [612,613,614,615], a heightened awareness of these relationships is important.

## 6. Potential Pathophysiological Mechanisms

A review and detailed discussion of multiple mechanisms underlying the potential association between HPI and OP/OFs is beyond the scope of the current manuscript, but a brief simplified re-count of most relevant factors might be useful. Proposed mechanisms and potential pathways linking numerous chronic HPI-associated diseases and disorders with OP/OFs are complex and include interplay between HPI-induced changes in gastroduodenal mucosal structure and function causing multitude disturbances in different biochemical pathways and affecting many system organs. These effects are interrelated and can be grouped as follows: (1) local and systemic low-grade inflammation; (2) disturbances in mineral homeostasis (calcium, phosphate and magnesium); (3) alterations in hormonal status (gastric/gut production of ghrelin, somatostatin, gastrin, histamine, leptin, estrogens, serotonin and dopamine as well as secretion of systemic hormones—sex hormones, PTH, cortisol, etc.); (4) iron, vitamin B12 and folate deficiencies and anemia; (5) oxidative stress; (6) gut dysbiosis; (7) altered energy homeostasis; (8) *H.p.* antigenic mimicry (homologous sequences of some bacterial virulence peptides that mimic host proteins cause production of autoantibodies); and (9) drug-induced (influences of various medications used for HPI-induced diseases and HPI-related conditions). Notable, no single factor could account for HPI-associated OP/OFs. HPI induces a chain of events resulting in co-influence of alterations in multiple (physio-) pathological pathways which directly and indirectly affect the musculoskeletal system, gait, balance, hemodynamic stability, etc., compromising bone health and increasing risk of falling. Figure 2 summarizes the potential pathophysiological mechanisms linking HPI-related gastroduodenal and extra-digestive diseases and disorders with bone health, falls and OFs.

## 7. Clinical Implications and Recommendations

Although colonization with *H.p.* is not a disease in itself, HPI is considered the most important etiological factor for developing main gastroduodenal disorders and a possible contributor to various extra-digestive conditions linked with OP/OF. HPI as a potential determinant of the stomach/gut–OP/OF axis remains largely overlooked and neglected. Although currently most physicians remained skeptical of HPI–OP/OF relationship, the emerging information indicates that the possibility of such link should not be dismissed. The practical rational and need to pay more attention to HPI (among other non-heritable and modifiable RF for OP/OFs) is further supported by the following facts. Firstly, the high level of genetic predisposition to OP, muscle status and OF: the heritability of BMD was estimated between 60% and 90% [616,617,618], the heritability of OFs was estimated between 9.5% [619] to 16–35% [620,621,622] and 53–70% [617,623,624] and the heritability of muscle-related traits was estimated between 30% and 50% [625,626,627]. Secondly, only in half of patients (between 30% and 50%) antiresorptive drugs are effective [628,629,630,631,632]. Thirdly, among lifestyle preventive measures, only muscle strengthening, balance and posture exercises are beneficial (level of evidence A), whereas reduced smoking (level of evidence C) and alcohol consumption (level of evidence D) demonstrate low relevance [633,634]. It seems reasonable that accounting for more modifiable factors which contribute to OP/OF would reveal new possible states and thus lead to better management.

The heterogeneousness of HPI–host interactions indicates that the understanding and interpretation of the HPI–OP/OF link(s) could not focus on a single organ disorder but should account the highly complex integrated processes involving different system organs. These processes may significantly differ in individual patients and the identification of predominant mechanisms as determinants of clinical outcome(s) might be crucial for personalized management. Each patient with HPI has their own interplay of different pathomechanisms which may impact skeleton, falls or both. General and HPI-related RFs for OP/OF are summarized in Table 3. Effective reduction of both bone loss and risk of falls is equally important to prevent OFs.

We propose a five-step algorithm to address and integrate the potential role of HPI in OP/OF management (Table 4). This practical approach includes, in addition to evaluation of common RFs, assessment for presence or history of HPI-associated diseases and concomitant treatments focusing on use of bone affecting and fall-risk increasing drugs (Step 1), and, when indicated, investigation the patient’s HPI status (Step 2), the main bone-mineral characteristics (Step 3) and the specific HPI-related conditions/complications (such as iron deficiency/anemia and gut dysbiosis) linked to OP/OF (Step 4), and introduction of a personalized and holistic therapeutic strategy addressing the identified disorders and their combinations (Step 5). The proposed algorithm is based on the existing clinical and pathophysiological data and appears well feasible. Such approach, we hope, may help to identify more high-risk patients and yield improved information on individualized prediction, prevention and treatment of OP/OF; it can be useful for early intervention strategies targeting novel potentially preventable and treatable conditions.

Noticeable, common RF favoring development of OP/OF are frequently present in patients with HPI (Table 3). Our ability to fully understand countless HPI effects and risks for any one patient is still limited. To illustrate some clinically relevant examples of up- or downregulated pathways that may eventually trigger OP/OF, we briefly describe the effects related to site-specific differences in HPI-induced gastritis (two of multiple possible scenarios) and the role of two HPI-induced disorders—iron deficiency/anemia and gut dysbiosis.

Scenario 1. In a patient with HPI-induced (especially with cagA+ strains) chronic/atrophic corpus gastritis, several factors may affect OP/OF. Hypo-/achlorhydria; iron and/or vitamin B12 insufficiency/deficiency; hypoghrelinemia; decreased production of gastric/gut estrogens and dopamine; increased gastrin, histamine and PTH production; alterations in serotonin levels; and gut dysbiosis may impair nutrients absorption, cause alterations in mineral metabolism (calcium, phosphate and magnesium) and stimulate bone degradation. These factors and the systemic low-grade inflammation may contribute to dysregulations in many other system organs and affect energy homeostasis resulting in muscle weakness, hemodynamic instability, weight loss and gait abnormalities—disorders associated with OP and elevated risk for falling.

Scenario 2. In individuals with HPI-induced predominantly antral gastritis, decreased somatostatin and increased gastrin levels, gastric acid hypersecretion and PUD (in a significant proportion of affected subjects) would be expected. In these patients, iron deficiency/anemia may be caused by gastroduodenal bleeding (a common PUD complication), while use of PPIs may affect absorption of calcium and some nutrients and in combination with HPI contribute to gut dysbiosis, a condition strongly associated with OP/OFs.

Crucially, in any case, an increased expression of systemic low-grade inflammation (due to microbe’s specific virulence factors), hosts’ genetic predisposition and/or microbe’s antigenic mimicry, alongside the abovementioned HPI-induced metabolic and hormonal alterations, might convey strong direct and indirect effects on bone homeostasis, muscles, cardiovascular, nervous and other systems linked with OFs.

Among the wide array of different diseases associated with both HPI and susceptibility to OP/OF, we would like to emphasize particularly the role of HPI-induced alterations in iron homeostasis and gut microbiota. Both these disorders are significantly implicated in a wide variety of disease states linked to OP/OFs but often overlooked and ignored in clinical practice.

Iron deficiency/anemia is a worldwide highly prevalent disorder (1.2 billion people [646]; 15–30% among older adults [647,648]) associated with detrimental effects in multiple medical conditions [649] including OP/OF. HPI is considered an important cause of iron deficiency and anemia; these conditions are among few disorders included in HPI international consensus management guidelines [360]. However, in persons with HPI-induced iron deficiency/anemia, diagnostic tests for OP and preventive osteoporotic treatment are still not advocated. The awareness that impaired iron status may be induced by HPI and may lead to OP/OF is critical. Individuals with OP/OF should be screened and monitored for iron deficiency/anemia, its underlying etiology identified (is it HPI-related?) and the iron status corrected. On the other hand, all clinicians, regardless of their specialty, should be encouraged to assess bone status in patients with iron deficiency/anemia and to introduce, when appropriate, osteoporotic therapy. Currently, however, clinical management (diagnosis and treatment) of iron deficiency is suboptimal or even inadequate [650], particularly in regard to OP/OF [358,651,652]. Building inter-disciplinary bridges to address this problem is important.

The other topic which deserves more attention is HPI-associated dysregulation of gut microbiota [239,606,653,654,655,656,657,658,659,660,661,662,663,664,665,666,667], a phenomenon described also in animal models (infected Mongolian gerbils [668]) and thought to be due mainly to gut immunopathological/inflammatory responses and suppressed gastric acid secretion following chronic/atrophic gastritis or use of acid-suppressive drugs. Gut dysbiosis by affecting various biological processes contributes to the pathogenesis of OP [669,670,671,672,673,674,675,676,677,678,679,680,681,682,683,684,685,686,687,688,689,690,691,692,693,694,695,696,697,698,699] as well as to numerous other extra-digestive system diseases associated with OFs [700,701,702,703,704,705,706,707,708,709].

Within this context, it should be mentioned that gastrointestinal microbiota profiles may also be altered by medications commonly used in HPI-related diseases (antibiotics, PPIs, SSRIs, opioids, NSAIDs, antipsychotics, anticholinergic inhalers, polypharmacy) [701,710,711,712,713,714,715,716,717,718,719,720,721]. The most recent publication found that 19 of 41 analyzed drugs influence gut microbiome composition [720]; PPIs, antibiotics, metformin, SSRIs, NSAIDs, antipsychotics and laxatives showed the strongest associations [715,717,720]. The PPI-induced dysbiosis has been reported in recent studies and reviews [715,718,722,723,724,725,726,727] and again confirmed in the latest systemic review based on 11 interventional and 12 observational cohort studies [728].

The eradication of HPI can restore the microbial diversity and increase the abundance of beneficial bacteria (e.g., Lactobacillus and Bifidobacterium) [729]. Restoring a balanced gut microbiota is considered as a therapeutic tool for various diseases, including OP [695,730,731,732,733,734,735,736,737]. Recent studies showed that probiotic and prebiotic supplementation increases bone density in healthy individuals, protects against primary and secondary OP [687,695,699,734,738,739,740] and exerts beneficial effects in many other extra-digestive diseases [740]. However, despite the established impact of the gut microbiota on host (patho-) physiology, the potential roles of HPI- and medications-induced gut dysbiosis is not mentioned in current guidelines for OP.

To put together, the challenge is to find the right practical approach to evaluate simultaneously presence of HPI and its characteristics, the bone status and risk of falls and timely introduce appropriate individualized therapies avoiding both under- and over-diagnosis. In patients with HPI-induced diseases bone characteristics, risk for falls and fractures should be assessed, whereas in individuals with risk or presence of OP/OF the investigation for HPI needs to be considered and if the infection identified appropriate treatment prescribed.

Since HPI is associated with many extra-digestive pathologies which might not be clinically obvious (silent alterations in mineral-bone metabolism, iron and/or vitamin B12 deficiencies, hormonal imbalance, gut dysbiosis, etc.) and patients are often followed by clinicians other than gastroenterologists (primary care doctors, cardiologists, hematologists, endocrinologists, oncologists, orthopedic surgeons, etc.), multi-disciplinary cooperation/collaboration is important and cross referrals should be considered. A multidisciplinary approach seems to be fruitful when dedicated professionals appreciate each other’s contributions and priorities.

Recognizing multiple HPI-induced diseases as contributing to both bone health and falls and deeper understanding the complex biology of the HPI–OF axis may lead to reconceptualizing the OP/OF management; the proposed algorithm, we expect, can be used as the first draft. In addition to existing strategies, the new interventions might include assessment the HPI status, HPI eradication, use of ghrelin (and its analogues), especially in HPI-induced chronic/atrophic corpus gastritis, treating (when appropriate) iron and vitamin B12 deficiencies, correcting gut dysbiosis—to mention some therapies which showed encouraging results and needed to be confirmed in longitudinal studies.

However, in recent international guidelines for prediction and prevention OP/OFs, the potential role of HPI is not mentioned [358]. Detection and eradication of *H.p.* are not part of OP/OF management, despite very high prevalence of HPI throughout the world (more in the elderly), growing research data on HPI–OP/OF link and the alarming expectations that in the coming decades the incidence of OFs will significantly increase worldwide as the population is ageing [741,742,743,744,745]. In short, taking a wider approach on OP/OF, considering the potential pathophysiological role(s) of HPI-induced diseases and disorders and generating a new paradigm may improve managing strategies, provide new angles for optimization OP care, prevent future fractures and lead to novel therapeutic options.

## 8. Limitations

It is worth emphasizing, as stated in the previous sections, that interpretation of controversial data on HPI–OP/OF relationships should take into account the methodological weaknesses most of the available studies (the vast majority were cross-sectional). The conflicting results are likely due not only to differences in design, sample sizes, protocol and methodologies of determination HPI (often limited to seroprevalence only) as well as OP (T-score BMD method underestimates the OF risk in 56–82% [746,747,748,749,750]), but also—and more importantly—very limited information on specific microbiological characteristics of H.p., ignoring predominant site and severity of gastritis, heterogeneity in patient populations (wide variations in demographic, socioeconomic, lifestyle, dietary and environmental confounding factors, comorbidities, risk of falling, used medications), However, despite these limitations, the huge number of publications reporting significant links between HPI, particularly with virulent strains (cagA+), and the development, course and severity of different chronic gastroduodenal and extra-digestive diseases and disorders known to be associated with falls, OP and, consequently, OFs cannot be ignored. The topic requires reconsideration in well-designed prospective studies examining simultaneously the virulence factors of *H.p.*, host’s susceptibility, its neuroendocrine–immunological-inflammatory response to HPI, socio-demographic, environmental and clinical characteristics including objective measurements of bone status (mineral-bone metabolism, BMD), risk factors for falls and occurrence of OF.

## 9. Conclusions

In this review, we attempt to illuminate the existing clinical information on links between HPI and OP/OFs and, the complexity and interdependence of HPI–host interactions. The available evidence indicates that diseases and disorders induced by HPI (especially with virulent strains (cagA+) may contribute directly and indirectly to the development and progression of OP, falls and OFs. Despite remaining gaps in knowledge (the underlying mechanisms have not been definitely proven), there is considerable amount of data to suggest that predictive, preventive and therapeutic strategies for OP/OFs should assume HPI-related pathologies as potential pathophysiological co-factors and concentrate on individualized management of their effects on both bone health and falls. In patients with HPI-associated diseases and disorders bone status and risk for falls and fractures should be assessed, whereas in individuals with risk or presence of OP/OF the HPI status needs to be investigated and appropriate treatment prescribed. A five-step algorithm to provide guidance on assessment of the possible contribution of HPI to OP/OF is presented; its clinical effectiveness needs to be validated. Further well-designed prospective studies are warranted to provide a deeper understanding of the HPI–OP/OFs axis and develop personalized preventive and curative therapies.

## 10. Key Points

*H.p*. colonizes about half of the world population. HPI as a multi-system condition confined not only to gastroduodenal morbidity but also many chronic extra-digestive diseases (CVD, neurodegenerative, endocrine, CLD, CKD, etc.) might directly and/or indirectly affect bone status, predispose to falls and, consequently, to OFs.The relationship between HPI and OP/OFs, two common, multifactorial and heterogeneous conditions, depends on complex interactions of multiple factors, including microbe’s virulence, host genetic predisposition, local gastroduodenal and systemic responses (biochemical, metabolic, hormonal, immunologic and inflammatory) and environmental influences. Therefore, microbe’s contribution to development and progression of OP/OF and the risk profile in colonized individuals could vary significantly. When studying the role of HPI in OP/OF, correction for the mentioned factors, is essential.The data on associations between HPI and OP/OFs in the literature are inconsistent, but there is growing evidence that HPI (especially in persons infected with virulent strains, e.g., *cagA+*) increases risk of OP/OF approximately 1.5–2-fold.Given the widespread prevalence of HPI in the population, the practical implication for these data is that comprehensive assessment for OP/OF risks should include evaluation for HPI-related diseases and disorders and vice versa (assessment for HPI in subjects with established OP, falls and low energy fractures); such approach would assist in individualized prevention and treatment of OP/OFs and should be considered at health care policy level.The usefulness and applicability of a practical strategy addressing HPI, an easy identifiable and treatable condition, as a potential pathophysiological co-factor of OP/OF, are worth further investigation in controlled, long-term studies with simultaneous assessment of *H.p*., host’s and environmental characteristics; a better understanding of the mechanisms underlying HPI–OP/OF relationship and individual outcomes should be achieved.

## Figures and Tables

**Figure 1 jcm-09-03253-f001:**
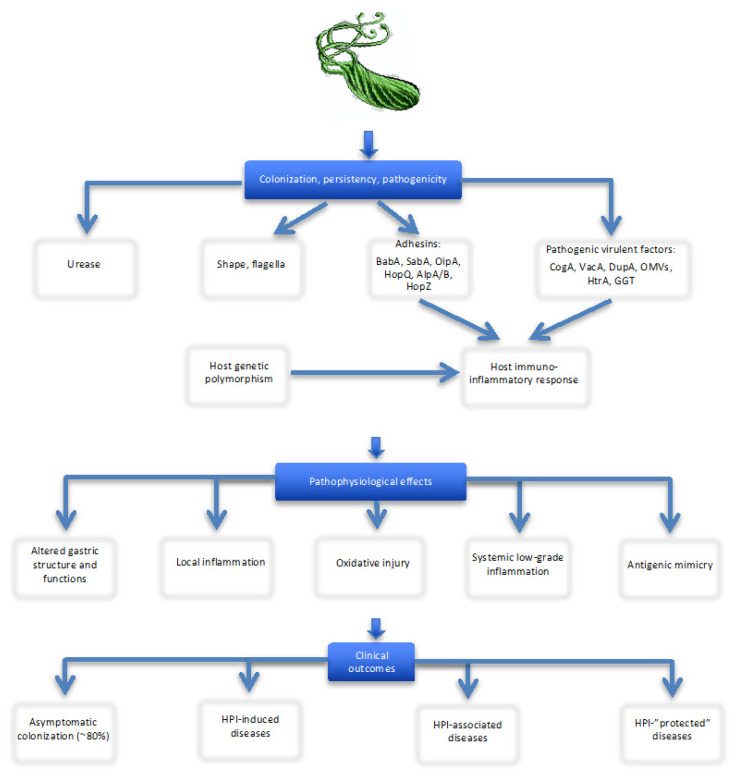
Overview of *Helicobacter pylori* (*H.p.*) characteristics contributing to gastric colonization, persistence, pathogenicity and clinical outcomes. Abbreviations: AlpA/B, adherence-associated lipoprotein A and B; BabA, blood group antigen-binding adhesin; CagA, cytotoxin associated antigen A; DupA, duodenal ulcer (DU) promoting antigen; GGT, gamma-glutamyl transpeptidase; HopQ, HopZ, *H. pylori* outer membrane proteins; HtrA, high-temperature requiring protein; OipA, outer inflammatory protein; OMVs, outer membrane vesicles; SabA, sialic acid-binding adhesin; VacA, vacuolating cytotoxin. Notes: The outcome of *H.p.* is multifactorial and depends on interaction between multiple heterogenic bacterial virulence factors, host genetics, lifestyle and environmental influences. *H.p.* utilizes a variety of mechanisms which allow: (1) escaping high acid environment (urease, bacterial shape and flagella); (2) attaching to the gastric epithelial layer (adhesin proteins); (3) exerting epithelial cell pathogenicity and (4) affecting the host innate and adaptive immune responses. The expression of virulence factors and host’s immunologic responses (dependent of genetic predisposition/resistance, e.g., proinflammatory cytokine gene polymorphisms) are critical to host colonization, infection persistence and pathogenesis of local (gastroduodenal) and systemic (extra-digestive) diseases. The cascade of pathophysiologic events in the stomach includes acid neutralization, mucus layer destruction, immune cell activation (lymphocytes, macrophages, dendritic cells, natural killer and mast cells), upregulation of pro-inflammatory (IL-1β, IL-6, IL-8, IL-17,TNF-α, IFN-γ and CRP) and anti-inflammatory (IL-4 and IL-10) cytokines (immune-inflammatory axes) and increased production of reactive oxygen species (oxidative stress) causing cell damage, alterations of gastric structure and functions (including changes in gastric acid and pepsin secretion, hormone production) as well as numerous effects on the gut (motility and microbiota) and extra-digestive organs; these may result in gastroduodenal erosion, peptic ulcer, carcinogenesis or lymphoma formation, as well as contribute to development and progression of numerous chronic diseases outside the stomach (CVDs, neurodegenerative, hematologic, metabolic, CKDs, CLDs, etc.); however, the role of HPI is not necessarily detrimental, it may even be “protective” (asthma in children; allergy; IBD, especially Crohn’s disease; and autoimmune disorders). Infection with virulent strains (in particular, cagA+ and vacA+) is associated with higher inflammatory response, oxidative injury and elevated risk of gastroduodenal and most extra-digestive diseases.

**Figure 2 jcm-09-03253-f002:**
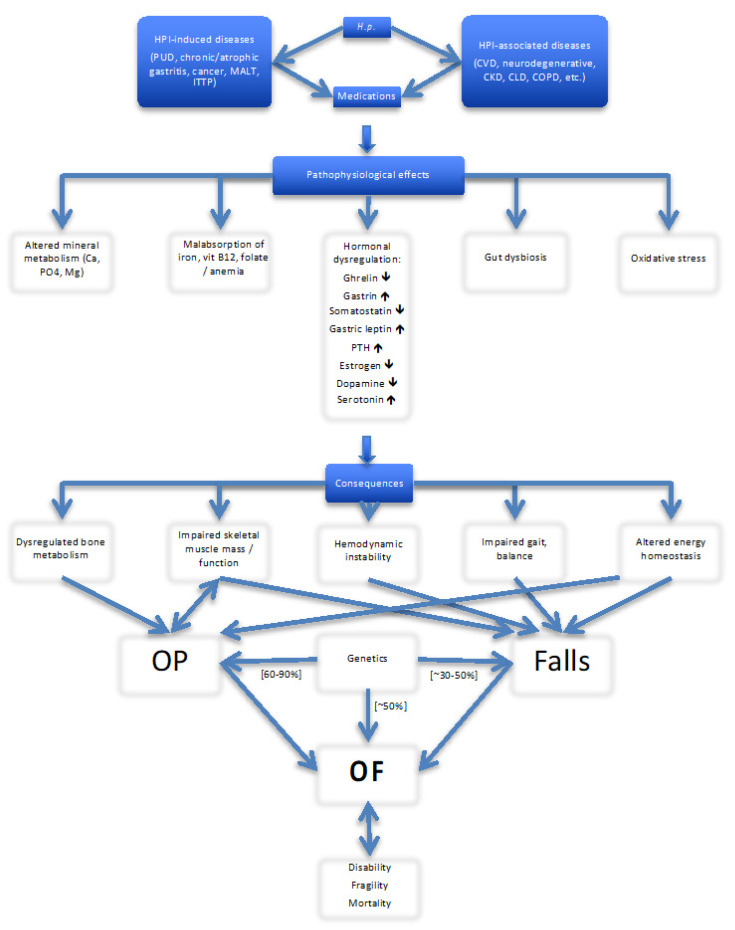
Schematic representation of Helicobacter pylori infection (HPI)-related factors and main pathophysiologic mechanisms that may lead to osteoporotic fractures. Abbreviations: Up-arrows refer to elevation/stimulation, down-arrows refer to reduction/decrease; Ca, calcium; CKD, chronic kidney disease; CLD, chronic liver disease; COPD, chronic obstructive pulmonary disease; CVD, cardiovascular disease; *H.p.*, *Helicobacter pylori*; ITT, idiopathic thrombocytopenic purpura; MALT, mucosa-associated lymphoid tissue B-cell lymphoma; Mg, magnesium; OF, osteoporotic fracture; OP, osteoporosis; PO_4,_ phosphate; PTH, parathyroid hormone; PUD, peptic ulcer disease. Notes: HPI- and concomitant treatment-induced changes in gastric structure and functions affect nutrient absorption (calcium, phosphate, magnesium, iron, vitamin B12, folate, etc.), production of gastric/gut (ghrelin, gastrin, histamine, somatostatin, leptin, estrogens, dopamine, serotonin and incretins) and systemic hormones (PTH and sex hormones) and cause gut dysbiosis. These alterations (via direct and/or indirect mechanisms) result in dysregulated bone-mineral metabolism, impaired skeletal muscle mass, function, gait and balance, hemodynamic instability and altered energy homeostasis increasing the risks for OP, falls and, consequently, OF, which ultimately lead to disability and mortality. Consequences of HPI and OP/OF can be connected and influence each other. In an individual patient, the contribution of HPI to development and progression of OP/OF is determined by a constellation of different but interdependent genetic, hormonal, metabolic, inflammatory and nutritional pathomechanisms, each of which alone and/or in combination may cause specific impairments of multiple organs (remote from the primary infection site) and their functions linked to OFs. The involved processes are highly complex, integrated and may significantly differ in individual patients. Of note, the estimated heritability of BMD is 60–90%, of muscle-related traits is 30–50%, and of OFs is approximately 50% (9.5–70%), indicating the importance to address non-heritable modifiable (including HPI-related) risk factors for OP/OF. (The roles of the site, type and severity of the HPI-induced gastritis and effects of specific treatments in the possible causative relationship between HPI and OP/OF are not depicted here).

**Table 1 jcm-09-03253-t001:** Data on relationship between *H. pylori* infection (HPI) and bone status/osteoporosis.

First Author, Year, Reference No, Country/Region	Population, Gender (F/M)	Mean Age, Years	HPI Detection Method	HPI Strain Identification	Bone Status Assessment Methods, Skeletal Location	HPI+ and OP, *n* (%)	HPI+ and Non -OP/Controls	Association (Yes/No)
Figura N, 2005 [98], Italy	240, M	65 (55–82)	Serum antibody	CagA	DEXA, LS, FN; Urinary cross-laps, serum bALP, PTH, Ca, PO_4_, 25(OH)D	51/80 (63.7%); CagA+: 30 (58.8%)	107/160 (66.8%); CagA+: 43 (40.1%)	Yes, if CagA+; increased levels of urinary cross-laps
Ozdem S, 2007 [105], Turkey	61, 36/25	11.8 ± 3 (F) 10.1 ± 3 (M)	RUT, histology		Serum P1NP, βCTX, OC, ALP, PTH, Ca, PO_4_			No
Kakehasi A, 2007 [107], Brazil	50, F	61.7 ± 7 (50–70)	RUT, histology, ^13^C-UBT		DEXA, LS	10/18 (55%)	24/32 (75%)	No
Kakehasi A, 2009 [108], Brazil	85, F	63.7 ± 7.3 (HPI+); 62.5 ± 7.0 (HPI−)	RUT, histology, ^13^C-UBT		DEXA, LS, H			No
Figura N, 2010 [99], Italy	1118, 935/183	62.5 ± 6 (F); 65.9 ± 6 (M)	Serum antibody	CagA	DEXA	41.5%; CagA + 30%	43.9%; CagA + 21%	Yes, if CagA+
Akkaya N, 2011 [109], Turkey	105, F	65.3 ± 6.1 (OP+) 63.6 ± 6.5 (OP−)	Serum antibodies		DEXA, LS, H	41/58 (IgG+) (70.7%)	35/47 (IgG+) (74.5%)	No
Asaoka D, 2014 [110], Japan	200, 105/95	62.8 ± 7.7 (M) 63.4 ± 9 (F)	Serum antibody, ^13^C-UBT		DEXA, LS; serum bALP, NTX	25/41 (61.0%)	57/159 (35.8%)	Yes, OR 5.33 (1.73–16.42) in PUD
Lin S, 2014 [111], Taiwan	365, F	77.3 (65–97)	RUT, histology		DEXA or osteoporosis medication use	77/101 (76.2%)	24/101 (23.8%)	Yes, OR 2.03 (1.14–3.62)
Asaoka D, 2015 [106], Japan	255, 135/120	63.2 ± 8.5	Serum antibody, ^13^C-UBT		DEXA, LS; serum bALP, NTX	25/43 (58.1%)	69/212 (32.5%)	Yes, OR 3.0 (1.31–6.88)
Mizuro S, 2015 [112], Japan	230, M	>50; 62.1 ± 5.0 (TBD low), 58.4 ± 5.4 (TBD normal)	Serum antibody		QUS, radius	61/116 (52.5%)	38/114 (33.3%)	Yes, OR 1.83 (1.04–3.21)
Fotouk-Kiai M, 2015 [113], Iran	967, 392/575; *H.p.*+ 758, *H.p.*− 209 (controls)	68.3 ± 6.8 (*H.p.*+) 69.3 ± 7.4 (*H.p.*−)	Serum antibody		DEXA, LS, FN	236/758 (31.1%)	522/758 (68.9%)	No, OR 0.76 (0.55–1.05)
Chung Y, 2015 [97], Korea	I, 126, M *H.p.*+ 657 *H.p.*− 469	54.4 ± 10.7 (H.p.+) 51.9 ± 12.1 (H.p.−)	Serum antibody		DEXA, LS (L1-L4)	173 */657 (26.3%) (LS); 114/657 (17.4%) (FN)	484/657 (73.7%) (LS); 543/657 (82.6) (FN)	Yes, only for lumbar BMD (not for total femur or femoral neck)
Kalantarhormozi M, 2016 [96], Iran	250, F; 16 (OP), 234 (controls)	58.9 ± 8.0	Serum antibody		DEXA, LS, F; bone turnover markers, OPG, RANK, Ca, PO_4_			No
Shih H, 2016 [95], Taiwan	5447 (*H.p.*+), 21,788 (controls)	>20	*H.p.* eradication treatment for PUD		DEXA			Yes, HR 1.62 (1.06–2.47)
Chen L, 2017 [114], Taiwan	2689, 1792/897	>40	^13^C-UBT		FRAX (without BMD)	F: 177/324↑ (54.6%); M: 54/93 (58.1%)		No, for 10-year fracture risk prediction
Chinda D, 2017 [115], Japan	473 F (healthy)	52.2 ± 15.2	Serum antibody (IgG), *H.p.* antigen in stool sample		QUS, calcaneus	65 */118 (55.1%)	53/118 (44.9%)	No, OR 0.95 (0.55–1.63) for osteopenia
Abdolahi N, 2017 [116], Iran	107 F, 34 with OP, 73 controls	Post- menopausal	Serum antibodies (IgA, IgG)			70.6% IgA+ 82.0% IgG+	54.8% IgA+ 75.3% IgG+	No
Lu L, 2018 [19], China	1867, 393/1474		^13^C-UBT		QUS, calcaneus			No, for BMD
Pan B, 2018 [117], Taiwan	867, 299/568	55.9 ± 11.3	RUT		DEXA	257/556 (46.2%)	124/311 (39.9%)	Yes, OR 1.62, (1.12–2.35) for decreased BMD
Chinda D, 2019 [118], Japan	268 M (healthy)	49.1 ± 15.1	Serum antibody (IgG), *H.p.* antigen in stool sample, serum pepsinogens		QUS, calcaneus			No, OR 1.31 (0.54–3.21) for atrophic gastritis, OR 0.74 (0.29–1.90) without gastritis

Abbreviations: ALP, alkaline phosphatase; bALP, bone-specific ALP; βCTX, β-collagen1 carboxy-terminal telopeptide; Ca, calcium; DEXA, dual energy X-ray absorption; CagA, cytotoxin associated antigen A; DM, type 2 diabetes mellitus; FN, femur neck; H, hip; H.p, H. pylori; LS, lumbar spine; HR, hazard ratio; LS, lumbar spine; NTX, collagen type-1cross-linked N-telopeptide; OC, N-mid-osteocalcin; crosLaps, urinary type 1 collagen C-telopeptides; OPG, osteoprotegerin; OR, odds ratio (in brackets 95% confidential intervals); P1NP, N-terminal cross-links of human procollagen type1; PO_4_, phosphate; PTH, parathyroid hormone; QUS, quantitative ultrasonic densitometry; RANK, receptor activator of nuclear factor κB; RUT, rapid urease test; 25(OH)D, 25-hydroxy vitamin D; Serum antibody, serum anti-H.pylori antibody (ELISA kits); TBD, trabecular bone density; UBT, urea breast test; ↑, high 10-year fracture risk using the Fracture Risk Assessment tool (FRAX) scale without BMD; *, osteoporosis and osteopenia combined. Empty cells indicate that data were not mentioned.

**Table 2 jcm-09-03253-t002:** HPI-related chronic extra-gastroduodenal diseases and disorders linked to osteoporotic fractures.

Iron deficiency and iron deficient anemia
Vitamin B12 deficiency and vitamin B12 deficient anemia
Immune thrombocytopenia
Cardiovascular diseases (CAD, myocardial infarction, hypertension, CHF)
Cerebrovascular diseases (stroke, TIA)
Neurodegenerative diseases (Alzheimer’s and vascular dementia, Parkinson’s disease)
Chronic kidney disease
Diabetes mellitus
Metabolic syndrome
Chronic liver disease (MAFLD; liver cirrhosis, hepatic encephalopathy)
Chronic obstructive pulmonary disease
Depression, anxiety
Rheumatologic and autoimmune diseases (rheumatoid arthritis [?], ankylosing spondylitis, psoriatic arthritis, systemic vasculitis, autoimmune thyroid diseases, multiple sclerosis [?])
Eye diseases (open-angle glaucoma, neuromyelitis optica)
Malignant tumors (breast, colorectal and prostate cancers)
Malnutrition/low body weight
Malabsorption
Vitamin D insufficiency/deficiency
Dysregulation of gastrointestinal microbiota (dysbiosis)
Chronic inflammatory bowel diseases [?]
Celiac disease [?]
Obesity (morbid) [?]
Prostatitis
Pre-eclampsia

[?] Indicates that HPI according to some studies may protect against the disease. Abbreviations: *CAD,* coronary artery disease; CHF, *congestive heart failure;* MAFLD, metabolic associated fatty liver disease.

**Table 3 jcm-09-03253-t003:** Risk factors (RF) for osteoporotic fracture in patients with *Helicobacter pylori* infection.

***General/Common RF***	***HPI-Induced Diseases and Disorders***
Advanced age	Anemia, iron deficiency
Menopause/male hypogonadism	Vitamin B12 deficiency
Body mass loss	Immune thrombocytopenia
Low BMD	Chronic/atrophic gastritis
Previous fragility fracture	PUD
History of falls	Gastric malignancy
Family history of OP/OFs	***HPI- associated chronic diseases and disorders***
Ethnicity (Caucasian and Asian vs. black populations)	Cardiovascular diseases (CAD, CHF, AF, hypertension)
Impaired balance, gait and mobility, need of assistive device *	Hemodynamic instability (orthostatic and/or postprandial hypotension, dizziness) #
Low physical activity/immobilization	Cerebrovascular diseases (stroke, TIA)
Low body mass index	Neurodegenerative diseases (dementia, Parkinson’s disease)
Hemodynamic instability (orthostatic and/or postprandial hypotension, dizziness) #	COPD
Visual impairment	CKD
Vitamin D deficiency/insufficiency	Diabetes mellitus
Vitamin K deficiency	Metabolic syndrome
Hyperparathyroidism	CLD
Urinary incontinence #	Depression, anxiety
Low calcium intake	Rheumatologic diseases
Fear of falling #	Eye diseases (open-angle glaucoma, neuromyelitis optica)
***Prolonged use of certain medications***	Gut dysbiosis
Corticosteroids, antidepressants (especially, SSRIs, SSNRIs), opioids, anxiolytics, hypnotics, sedatives (benzodiazepines), antiparkinsonian (dopaminergic) medications, antipsychotics, antiepileptics, glitazones, antiarrhythmics, PPIs, thyroxine, aromatase inhibitors, gonadotropin releasing hormone antagonists, immunosuppressive agents, polypharmacy	Malignant tumors (breast, lung, colorectal, prostate cancers)
***Environmental, lifestyle and socio-economic RF***	
Cigarette smoking, excess alcohol consumption, diet, urbanization, poor sanitation conditions, air pollution.	

Abbreviations: AF, *atrial fibrillation; CAD*, coronary artery disease; CHF, *congestive heart failure;* CKD, chronic kidney disease; CLD, chronic liver disease; COPD, chronic obstructive pulmonary disease; CVD, cardiovascular disease; PPI, proton pump inhibitor; PUD, peptic ulcer disease; SSRI, selective serotonin reuptake inhibitors; SSNRI, selective serotonin-norepinephrine reuptake inhibitors; TIA, transient ischemic attack; # Risk factor for falls (not for OP); * although need for an assistive device indicates presence of conditions predisposing to falls, its appropriate use may actually decrease fall risk. Notes: Recent genetic studies have challenged some long-assumed risk factors for OP/OF. Mendelian randomization analyses identified BMD [413,635,636,637], serum estradiol concentrations (in men) [638] and cigarette smoking [639] as causal risk factors for OP/OFs, whereas genetic predisposition to lower levels of vitamin D and milk calcium intake [635,636,639,640], serum testosterone [638] and inflammation markers [641,642], as well as early menopause; late puberty, chronic (including CVD, DM and IBD) [413,414,643] and neuropsychiatric diseases (Alzheimer’s disease, schizophrenia and bipolar disorder) [644], alcohol consumption [645] and alcohol dependence [639] did not show causal effects on BMD and fracture risk. The genetic studies overcome many limitations of the previous observational studies but also contain potential bias; “the Mendelian randomization study design cannot be used to assess whether complications or treatment of those diseases influence fracture risk” [636].

**Table 4 jcm-09-03253-t004:** Algorithm for osteoporotic fracture risk assessment and management in regard to *Helicobacter pylori* infection.

**Step 1.** Assess, in addition to evaluation common risk factors for OP/OF, presence or history of HPI-induced and HPI-associated diseases and disorders and check the appropriateness of concomitant treatments (the potential role of drugs used in regard to OP and falls) (see Table 3).
**Step 2.** If indicated, assess current HPI status, the microbe’s virulent characteristics, predominant site, type and severity of gastritis.
**Step 3.** Assess the bone-mineral status (BMD, bone turnover markers, serum levels of vitamin D, vitamin B12, PTH, calcium, phosphate and magnesium).
**Step 4.** Evaluate for and address HPI-related specific conditions/complications associated with OP and/or falls (e.g., iron deficiency/anemia, gut dysbiosis, hemodynamic instability, gait disturbance, frailty, etc.).
**Step 5.** Introduce a personalized and holistic care plan of preventive and/or therapeutic management according the identified disorders and their combinations. This may include: (1) HPI eradication; (2) adequate antiosteoporotic treatment; (3) elimination/minimization falls-related factors (alleviation effects of chronic diseases and/or drugs causing hemodynamic instability, gait disturbances, muscle loss, frailty); (4) review and optimization of all medications used; (5) correction the metabolic alterations (vitamin D, iron and vitamin B12), hormonal status, anemia and gut dysbiosis (e.g., pre- and probiotic therapy); (6) nutritional support; and (7) modulation of lifestyle factors (physical activity, tobacco smoking and alcohol consumption).

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
