# Peer review of "Helicobacter pylori Related Diseases and Osteoporotic Fractures (Narrative Review)"

_jcm, 2020, doi:10.3390/jcm9103253_

Round 1

Reviewer 1 Report

This is a very attractive thema and the authors devoted a lot of time collecting a great amount of relevant literature. The manuscript is well written and organized. I would suggest the following suggestions for further improvement

  1. I would suggest to use italics for scientific names of microbes e.g. Helicobacter pylori. I would also use the same abbreviation for Helicobacter pylori and its infection. E.g. Hp and Hp-I
  2. Figure 1 has to be further edited. Hp icon is of a low quality with visible pixels. Furthermore, if copyright exists, should be mentioned with theaccompanied allowance to be reproduced. The “bubbles” with are too close to arrows, some distancing might be optically required
  3. You may also want to add, that Hp has been histologically isolated in speciments from trabeculectomy
  4. Although the review is a narrative review (as the title claims), somehow report of systemic character is present in the manuscript with number of studies and design etc. Athough it is not false, it is not the classic-„textbook“ review. The tables with the relevant studies also look-like a systematic review. If this was the real intention I suggest to reconstruct the review by using at least the well established PRISMA guidelines
  5. I would like to read something regarding Vit-D receptors and Hp. There is relevant bibliography and it is of great interest
  6. Molecular Mimicry has been proposed further for Glaucoma and other pathologies.
  7. The protective theory of Hp on GERD is outdated. I highly recommend to revise this paragraph. Hp contributes actually to GERD exacerbation
  8. Consider using the new proposed term of NAFLD --> MAFLD (metabolic(dysfunction) associated fatty liver disease)
  9. I suggest to remove this protective paragraph or formulate it others -radically, after reading carefully a couple of papers of the Professor David Graham, one of the most important and prominent Helicobacter researchers world-wide

Doi: 10.1016/S0140-6736(05)66278-2, 10.1023/a:1022815902105

  1. When refering to vitamins and their levels we have to be precise what we mean and measure e.g. 25-OH-VitD, or Holocabalamin etc. Please adjust the text respectively
  2. Limitations: authors have to choose, if they want a pure narrative or a pure systematic review (where limitations really indicated) and adjust, respectively.

Author Response

Responses to the Reviewers*

To Reviewer 1

We are grateful for your time reviewing our manuscript and the conclusion that “This is a very attractive thema and the authors devoted a lot of time collecting a great amount of relevant literature. The manuscript is well written and organized”. We would like to thank you for your helpful comments, clarify some issues and present the changes which have been made.

Comment 1. I would suggest to use italics for scientific names of microbes e.g. Helicobacter pylori. I would also use the same abbreviation for Helicobacter pylori and its infection. E.g. Hp and Hp-I

Following your advice the name of the microbe (Helicobacter pylori) and its abbreviation (H.p.) is now shown in italics;the abbreviation HPI for Helicobacter pylori infection seems less confusing and more suitable.

Comment 2. Figure 1 has to be further edited. Hp icon is of a low quality with visible pixels. Furthermore, if copyright exists, should be mentioned with the accompanied allowance to be reproduced. The “bubbles” with are too close to arrows, some distancing might be optically required

Thank you, changes made.

Comment 3. You may also want to add, that Hp has been histologically isolated in speciments from trabeculectomy

Information on the presence of H.p. in trabeculum and iris included (p. 8, line 257; reference 86)

Comment 4. Although the review is a narrative review (as the title claims), somehow report of systemic character is present in the manuscript with number of studies and design etc. Although it is not false, it is not the classic-„textbook“ review. The tables with the relevant studies also look-like a systematic review. If this was the real intention I suggest to reconstruct the review by using at least the well established PRISMA guidelines.

We agree that some aspects our paper resemble a systematic literature review (clear and well-defined questions, quantitative and qualitative analyses, about 750 references, etc.); however, it does not include all published and unpublished studies on the topics discussed (currently PubMed contains >45.000 studies on H.p.). We did not present explicit description of inclusion and exclusion criteria for the reviewed studies and did not assess (but characterised) their methodology. Systematic reviews usually focus on and address one-two specific questions and often perform meta-analysis resulting in a pooled estimate of intervention effectiveness. Our paper, in contrast, aimed to obtain a broad perspective on the relationships between HPI and OP/OF, highlight the causes for controversies in the publications, and present a practical approach to management HPI in regard to OP/OF. Therefore, the review should be classified as a narrative review.

Comment 5.  I would like to read something regarding Vit-D receptors and Hp. There is relevant bibliography and it is of great interest

This is indeed a very interesting factor in pathogenic (especially immunologic) mechanisms involved in H.p. survival. However, our review focuses mainly on clinical aspects of the relationship between H.p. and OP/OF, and the complex underlying mechanisms are only listed (p. 28). The details of interactions between HPI and vitamin D, an essential antimicrobial defense factor, as well as an important contributor to musculoskeletal health (among multiple other beneficial effects), are beyond the scope of this article. This intriguing topic requires and deserves a separate review.

Comment 6.  Molecular Mimicry has been proposed further for Glaucoma and other pathologies.

The important pathophysiologic role of H.p. antigenic mimicry in different HPI-related diseases and disorders (including glaucoma [p.24, line 805) linked to OP/OF has been mentioned five times (p.6, line 172; p. 17, line 502; p. 25, line 814; p. 28, line 964; p.34, line 1118).

Comment 7.The protective theory of Hp on GERD is outdated. I highly recommend to revise this paragraph. Hp contributes actually to GERD exacerbation

As reviewers we attempted to present the published studies objectively highlighting the existing controversies on each topic; we fully followed this approach in summarising the information on GERD. We emphasised that the type and location of HPI-induced gastritis (corpus vs. antral), and accordingly the level of gastric acid secretion, should be taken into account as an important, if not the main, contributor to the HPI – GERD relationship, this may help the understanding and interpretation of the controversial reports.

Comment 8. Consider using the new proposed term of NAFLD --> MAFLD (metabolic (dysfunction) associated fatty liver disease)

Although we were aware about the recent international consensus (May-July 2020) regarding the new definition (MAFLD) instead of previous NAFLD, we did not use it as did other authors who published their papers in March-August in such prestigious Journals as JAMA, Nat Rev Gastroenterol., Hepatology. Now, however, the audience accepting the new definition is increasing, and it appears reasonable to use the new term. To avoid any confusion to the reader, a practicing physician in particular, we included both the new and old terms in the list of abbreviations: MAFLD/previous term NAFLD.

Changes:

p.23, line 752: metabolic associated fatty liver disease [MAFLD, previous term non-alcoholic…]

p.24, line 780: MAFLD;

p.24, line 807: MAFLD;

  1. 39, line 1304.

Comment 9. I suggest to remove this protective paragraph or formulate it others -radically, after reading carefully a couple of papers of the Professor David Graham, one of the most important and prominent Helicobacter researchers world-wide

We appreciate the work of Professor David Graham very much. Ten of his articles (reference numbers: 10, 43, 52, 55, 56, 62, 219, 241, 260, 360) including the one against the “protective” effect of HPI (number 241) are cited in our review. However, we do not think that it will be appropriate in a review paper to ignore the opposite research-based opinion repeatedly expressed in many publications including the most recent (up to 2019, numbers: 9, 78, 342, 433 – 442).

Comment 10. When refering to vitamins and their levels we have to be precise what we mean and measure e.g. 25-OH-VitD, or Holocabalamin etc. Please adjust the text respectively

In the cited studies the laboratory assessment of vitamin D status was, as usual, based on circulating 25 (OH) vitamin D levels [mentioned on p. 15, line 389]. For the detection of vitamin B12 deficiency most often serum B12 and rarely homocysteine concentrations have been measured, although holotranscobolamin level has been shown to be a moderately more reliable marker. To our knowledge, no studies on HPI measured simultaneously vitamin B12, holotranscobolamin, methylmalonic acid and homocysteine to assess vitamin B12 deficiency.

Comment 11. Limitations: authors have to choose, if they want a pure narrative or a pure systematic review (where limitations really indicated) and adjust, respectively.

We have addressed this question in our response to comment 4.

*All Reviewers comments are shown in original spelling.

Reviewer 2 Report

Thank you for the opportunity to review your manuscript. This review covers a controversial topic regarding the association between Helicobacter pylori and osteoporosis.

The authors provide a comprehensive review of H. pylori and potential mechanisms of of osteoporosis. While the review is thorough, there are areas in which weak associations between H. pylori and a variety of conditions such as orthostatic hypotension are discussed with additional information regarding medications (Section 5.3) that are not particularly relevant to H. pylori. The authors do a good job of investigating whether peptic ulcer disease or atrophic gastritis mediate the relationship between H. pylori and osteoporosis.

Similarly, the manuscript diverges from H. pylori by including a variety of risk factors for fragility fractures related to falls in table 3. For example, H. pylori cannot account for fear of falling in Table 3.

The authors can provide some clarifying information in their table 1. It would be helpful to include the odds ratio and 95% confidence interval if available in the Association (yes/no) column. 

There are also a few typographical errors that require revision:

Page 7, line 239: "Chief sells" should be "Chief cells"

Page 10, line 342: p=0.0.063 should be corrected

Page 12, table 1: "lumber" should be "lumbar"

Page 14: "lumber" should be "lumbar"

Figure 2. Make sure to spell check for words like “ghrelin”

Other than these minor issues, the authors should be congratulated on their extensive summary of the literature surrounding H. pylori and osteoporosis.

Author Response

Responses to the Reviewers*

To Reviewer 2

We are very thankful for your time reviewing our manuscript and gratified with your statement that “ Other than these minor issues, the authors should be congratulated on their extensive summary of the literature surrounding H. pylori and osteoporosis”.

We would like to clarify some issues and present the changes we made following your recommendations.

Comment 1. The authors provide a comprehensive review of H. pylori and potential mechanisms of of osteoporosis. While the review is thorough, there are areas in which weak associations between H. pylori and a variety of conditions such as orthostatic hypotension are discussed with additional information regarding medications (Section 5.3) that are not particularly relevant to H. pylori. The authors do a good job of investigating whether peptic ulcer disease or atrophic gastritis mediate the relationship between H. pylori and osteoporosis.

Indeed, on the first glance it appears that H. pylori is not associated with hemodynamic instability including orthostatic hypotension, which is known to be multifactorial. However, haemodynamic dysregulation, a well-established risk factor for falls and fractures, may be caused by numerous HPI-related diseases such as gastro-intestinal (especially when accompanied with inflammation and/or anaemia), CVDs (CAD, post-stroke, CHF, arrhythmias), T2DM (autonomic neuropathy, hypoglycaemia), neurodegenerative diseases (dementia, Parkinson’s), etc. Therefore, it is reasonable to emphasise and bring to reader’s attention the potential chain of events: HPI-related diseases and disorders - medication used - haemodynamic instability – falls and fractures).

Comment 2. Similarly, the manuscript diverges from H. pylori by including a variety of risk factors for fragility fractures related to falls in table 3. For example, H. pylori cannot account for fear of falling in Table 3.

No doubt that “H. pylori cannot account for fear of falling”, but this specific risk factor for falls is quite common in individuals (particularly in the elderly) with different HPI-related diseases and may lead to prescribing benzodiazepines, antidepressants and even antipsychotics, all of which significantly increase risk of falling and negatively affect bone metabolism. Following a holistic approach and aiming to prevent such cascade of events we included this risk factor in the summarising Table3.

Comment 3. The authors can provide some clarifying information in their table 1. It would be helpful to include the odds ratio and 95% confidence interval if available in the Association (yes/no) column. 

The available [in 9 studies] information on OR and 95%CI is already included in Table 1.

Comment 4. There are also a few typographical errors that require revision:

Page 7, line 239: "Chief sells" should be "Chief cells"

Page 10, line 342: p=0.0.063 should be corrected    

Page 12, table 1: "lumber" should be "lumbar”

Page 14: "lumber" should be "lumbar"

Figure 2. Make sure to spell check for words like “ghrelin”

Thank you very much for indicating the misspellings, all typos corrected.

*All Reviewers' comments are shown in original spelling.

Round 2

Reviewer 1 Report

In this revised manuscript, the review seems to be better shaped and more balanced. Authors have adequately addressed the majority of requested issues. I have no further comments